# SG-Gaze: Structurally and Geometrically Consistent Representation Learning for Generalizable 3D Gaze Estimation

## Abstract

Learning accurate and generalizable 3D gaze representations remains challenging due to the lack of a unified and physically grounded representation. Existing methods rely solely on appearance cues or simplified geometric modeling, and thus fail to jointly capture geometric and structural consistency. They exhibit poor cross-domain generalization and typically require large-scale multiview datasets to mitigate viewpoint variation, yet still struggle with domain gap between controlled and in-the-wild settings. To address these issues, we propose **SG-Gaze**, a dual-branch framework that learns a **S**tructurally and **G**eometrically Consistent **R**epresentation **(SGR)** for gaze estimation. The analytical branch embeds features onto a geodesically aligned spherical manifold for interpretable regression, while the model-guided branch reconstructs 3D eyeball structure with weak 2D edge supervision. Through adversarial alignment, the resulting SGR is simultaneously appearance discriminative, structurally faithful, and geometrically consistent. To further improve robustness, we introduce View-Consistent Regularization, which augments training SGR with synthetic view perturbations and enforces rotation-equivariant consistency across gaze vectors and structural projections. This reduces reliance on costly multiview data and mitigates cross-domain distribution shifts. Extensive experiments on synthetic and real-world datasets show that SG-Gaze achieves state-of-the-art accuracy and strong cross-domain generalization in 12 challenging transfer settings. Our work demonstrates the importance of unifying structural and geometrical consistency with equivariant regularization, providing broader insight into building more interpretable and generalizable models.

## 1 Introduction

Eye gaze estimation has broad applications in human–computer interaction (Admoni & Scassellati, 2017; Terzioğlu et al., 2020), driver monitoring (Cañas et al., 2025; Cheng et al., 2024), and immersive AR/VR systems (Burova et al., 2020; Konrad et al., 2020). Beyond practical use, it offers a unique perspective on addressing fundamental challenges in representation learning. The core problem of accurate 3D gaze estimation reflects a broader question: how to construct interpretable and physically grounded representations that generalize across viewpoints, devices, and domains.

Despite rapid advances, existing methods still face two key limitations: (1) **Lack of unified physical representation.** As shown in Fig. 1, appearance-based methods (Zhang et al., 2015; Chen & Shi, 2018; Cheng et al., 2020b) directly regress gaze from image features without explicit geometric constraints [Fig. 1(a)], while model-based methods (Chen et al., 2008; Hennessey et al., 2006; Świrski & Dodgson, 2013) reconstruct eyeball geometry but require personal calibration and dedicated devices [Fig. 1(b)]. These two isolated methods fail to combine geometric constraints with structural modeling to form a unified representation, resulting in weak interpretability and poor generalization across subjects and devices. (2) **Limited generalization across viewpoints and domains.** Gaze estimation models must be robust to diverse head poses, camera viewpoints, and environments. Existing models are sensitive to such changes. While they rely on costly multiview training data, but still struggle to bridge controlled and in-the-wild domains gap [Fig. 1(f)] due to the different perspectives distribution. This limitation stems from the lack of rotation-equivariant and geometric–structural consistent representations—two key properties for generalizable gaze modeling.

To address these challenges, we propose **SG-Gaze**, a dual-branch framework that learns a structurally and geometrically consistent representation **(SGR)** for 3D gaze estimation. Unlike traditional methods that focus solely on appearance or model, SGR unifies physical priors with representation learning to achieve both interpretability and generalization. Given a sequence of near-eye images,

Figure 1: Overview of gaze representation learning methods. (a) Appearance-based methods lack geometric constraints and interpretability. (b) Model-based methods require dedicated devices. (c) Our SG-Gaze unifies both methods by learning the most geometric(**G**) and the most structural(**S**) features grounded in gaze physics. (d) Adversarial learning fuses features into a unified Structurally and Geometrically Consistent Representation (SGR). (e-g) VCR enforces rotation-equivariant consistency to cover a larger field-of-view to mitigate domain gaps compared to GT supervision alone.

a shared backbone extracts discriminative visual features **F**. These features are then processed by two cooperative branches. The **A**nalytical **G**aze **E**stimation (**AGE**) branch encodes features onto a geodesically aligned spherical manifold [Fig. 1(c1)], explicitly enforcing the geometric constraint that gaze directions reside on a unit sphere. The **M**odel-**G**uided **R**econstruction (**MGR**) branch reconstructs 3D eyeball structures under weak 2D edge supervision [Fig. 1(c2)], enforcing structural anatomical constraint. A dedicated adversarial alignment module bridges the two branches [Fig. 1(d)], producing a unified SGR that preserves appearance cues while remaining consistent through geometric and structural priors. To further improve generalization across unseen viewpoints, we introduce **V**iew-**C**onsistent **R**egularization (**VCR**), which augments SGR with synthetic viewpoint perturbations and enforces rotation-equivariant consistency across gaze vectors and structural projections [Fig. 1(e)]. This physical constraint simulates multi-view conditions without costly data collection, mitigating controlled-to-real domain gaps by covering a larger field-of-view [Fig. 1(g)]. Finally, SG-Gaze jointly predicts 3D eyeball, gaze direction and auxiliary 2D semantic projections.

Comprehensive experiments on synthetic dataset UnityEyes (Wood et al., 2016) and real-world datasets TEyeD (Fuhl et al., 2021), LPW (Tonsen et al., 2016) show that SG-Gaze achieves state-of-the-art accuracy and strong cross-dataset generalization. We quantitatively validate its physical grounding through spherical correlation ($r = 0.89$), anatomical accuracy (e.g., $11.8 \pm 0.9$ mm eyeball radius), and low reprojection error(1.88 px). In summary, our main contributions are three-fold:

- We propose SG-Gaze, a unified dual-branch framework. By combining AGE and MGR based on the physical principles of gaze, SG-Gaze learns a Structurally and Geometrically Consistent Representation (SGR), which is interpretable and physically grounding.

- We introduce View-Consistent Regularization (VCR), a training strategy that enforces rotation-equivariant consistency of gaze vector and structure projection under synthetic viewpoint perturbations, alleviating domain gaps in a principled way.

- Experimental results show that SG-Gaze achieves state-of-the-art performance, with up to 38.61% baseline improvement across 12 challenging transfer scenarios without touching target domain data, backed by quantitative evidence of its physical plausibility.

## 2 MOTIVATION: STRUCTURAL AND GEOMETRIC CONSISTENCY

Our representation of gaze is grounded in two fundamental physical properties of human vision:
**Fact 1: Gaze vectors lie on a unit 3D sphere** Gaze direction can be defined as a unit vector from the eyeball center to a fixation point. Hence, all gaze vectors reside on the surface of a unit sphere $\mathbb{S}^2$. As shown in Fig. 2(b), the geodesic distance between two points on this sphere corresponds to the angular difference between gaze vectors, defining a structured manifold for gaze representation.
**Fact 2: Eyeball as an anatomically constrained structure** The eyeball approximates a rigid 3D sphere, where the gaze direction is determined by the geometric relation between the eyeball center and the pupil center. This means that accurate gaze estimation should not rely solely on image pixels or implicit features, but must respect anatomical plausibility and strong structural prior.

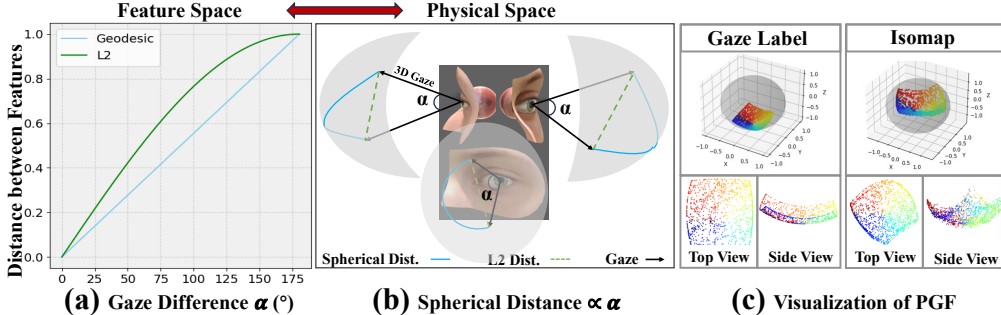

Figure 2: Physical geometry analysis of gaze features. (a) L2 and geodesic distances vs. angular differences on UnityEyes. (b) Geodesic distance on feature sphere proportional to gaze differences. (c) Visualization of the principal gaze feature after Isomap projection, where features exhibit a spherical distribution consistent with gaze labels, validating the geometric consistency of our representation.

**Spherical Structure in Feature Space** To examine whether feature space preserves geometric structure of gaze, we conduct experiments on UnityEyes (Wood et al., 2016). Source domain is denoted as $D_s = \{x_i, y_i\}_{i=1}^N$, where $x_i$ is eye image and $y_i$ the corresponding gaze vector. We pretrain a ResNet-18 (He et al., 2016) feature extractor $F_{\theta_1}(\cdot)$ with $L_1$ loss: $f_i = F_{\theta_1}(x_i)$, $\min_{\theta_1, \theta_2} \sum_{i=1}^N L_1\left(y_i, L_{\theta_2}(f_i)\right)$, where $f_i \in \mathbb{R}^{512}$ and $L_{\theta_2}(\cdot)$ is the final fully connected layer. We compute the geodesic distance between feature embeddings and compare it with angular difference of their ground-truth gaze vectors. **Key Observation**: The two exhibit a strong linear correlation (Pearson's $r = 0.89$), indicating that the feature space encodes the spherical topology of gaze directions. Motivated by this, we follow AGG (Bao & Lu, 2024) and extract a low-dimensional *Principal Gaze Feature (PGF)* by applying Isometric Mapping (Isomap) (Tenenbaum et al., 2000) to project $f_i$ into a 3D subspace. As shown in Fig. 2(c). The PGFs lie on the surface of a 3D sphere, preserving the structural geometry of gaze.

These complementary physical principles motivate reformulating gaze estimation as a representation learning problem under explicit structural and geometric constraints. We advocate learning a unified **Structurally and Geometrically Consistent Representation (SGR)** that simultaneously satisfies both constraints, enabling interpretable and robust gaze estimation across domains.

## 3 METHODOLOGY

We define the Structurally and Geometrically Consistent Gaze Representation SGR $\in \mathbb{R}^d$ that satisfies three key physical constraints.

**(1) Geometric Consistency**: SGR's 3D projection must lie on the unit sphere ($\|\text{SGR}_{\text{proj}}\| = 1$), preserving geodesic locality in $\mathbb{S}^2$ and the angular structure of gaze space: $d_g(\mathbf{z}_i, \mathbf{z}_j) \propto \angle(\mathbf{g}_i, \mathbf{g}_j)$.
**(2) Structural Consistency**: SGR must enable 3D eyeball reconstruction through a differentiable mapping: $\|\text{Proj}(D(\mathbf{z})) - \text{Edge}(I)\|_2^2$, where $D(\cdot)$ decodes 3D eyeball parameters and $\text{Proj}(\cdot)$ projects the 3D structure to 2D image, ensuring it corresponds to physically possible eyeball configurations.
**(3) Rotation Equivariance**: SGR must be consistent under 3D viewpoint changes: $D(T_P(\text{SGR})) \approx P \cdot D(\text{SGR})$, where $T_P$ is a learned rotation operator that approximates the physical 3D rotation P.

These constraints enforce that SGR not only predicts gaze direction but also maintains anatomical fidelity and geometric robustness, forming a unified and physically grounded representation.

### 3.1 OVERVIEW

The overall pipeline of SG-Gaze is illustrated in Fig. 3. Given a sequence of near-eye image frames $I = \{I_1, \ldots, I_n\}$, our framework estimates the 3D eyeball structure $e^m$, gaze direction $e^g$, and 2D semantics $e^s$ through a unified function $\Phi$, i.e., $\{e^m, e^g, e^s\} = \Phi(I)$. The framework consists of three core components: **(1) Base Branch:** Input frames $I$ are encoded by a shared feature extractor $B$ (ResNet-18) into visual features $F = B(I)$ [Fig. 3(a)]. **(2) Dual-Branch Decoding with Adversarial Learning:** The decoder $D$ contains two cooperative branches: the Analytical Gaze Estimation **(AGE)** branch [Fig. 3(b)] projects features onto a geodesically aligned spherical manifold to yield an interpretable embedding for gaze regression; the Model-Guided Reconstruction **(MGR)** branch [Fig. 3(c)] reconstructs 3D eyeball parameters (iris, pupil, sphere center) and jointly optimizes the gaze via physical modeling constraints through 3DEyeNet. Both branches share intermediate representation and are optimized jointly. A cross-task adversarial discriminator in $\Phi_{\text{afr}}$ [Fig. 3(d)] en-

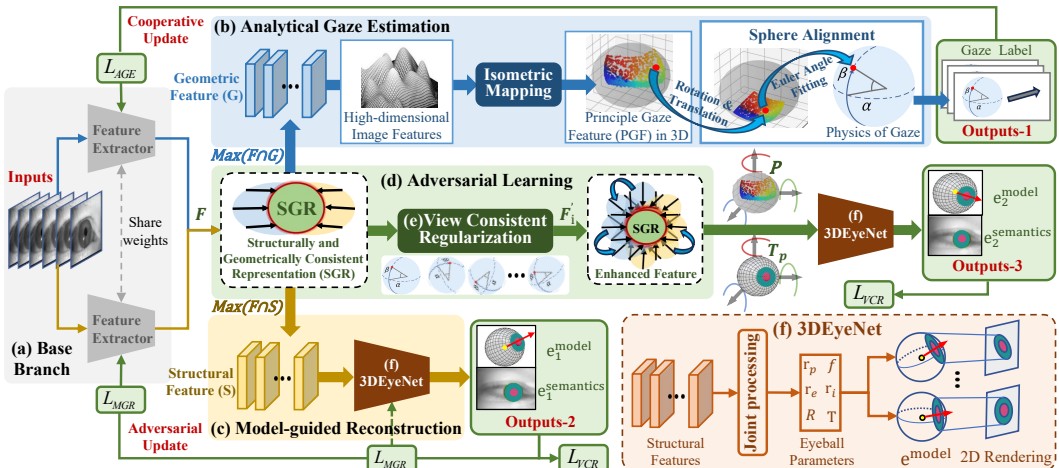

Figure 3: Overview of SG-Gaze. (a) Base branch: A shared feature extractor extracts visual features F. (b) AGE branch: Projects features onto a spherical manifold for geometrically consistent gaze regression. (c) MGR branch: Reconstructs 3D eyeball parameters under weak 2D edge supervision. (d) Adversarial learning: Ensures geometric and structural consistency through a discriminator network. (e) VCR module: Enforces rotation-equivariance with synthetic view-point perturbations. (f) 3DEyeNet: Implements differentiable rendering for anatomically plausible eyeball modeling.

courages AGE and MGR to produce structurally and geometrically consistent representation (SGR). **(3) View-Consistent Regularization (VCR):** In addition, the VCR module [Fig. 3(e)] applies synthetic rotations and enforces rotation-equivariant consistency of SGR across gaze vectors and structural projections, enhancing robustness to viewpoint variation without requiring multi-view data. Through the joint optimization of these components, SG-Gaze learns a unified SGR that seamlessly integrates appearance cues with physical modeling. It outputs the optimized 3D eyeball model $e^m$, gaze vector $e^g$, and auxiliary semantic projections $e^s$ with improved accuracy and interpretability.

## 3.2 ANALYTICAL GAZE ESTIMATION

The Analytical Gaze Estimation (AGE) branch explicitly encodes the physical geometry of gaze directions into the learning process. Unlike black-box regression, AGE leverages spherical projection and analytical fitting to construct gaze vectors that inherently respect the 3D spherical topology.

**Spherical Projection** As observed in Section 2, feature geodesic distances are linearly correlated with gaze angular differences. To preserve this topology, we apply Isomap to embed features $\mathbf{f}_i \in \mathbb{R}^{512}$ into a 3D spherical space: $\mathbf{p}_i\}_{i=1}^{N'} = \text{Isomap}(\{f_i\}_{i=1}^{N'})$, where $p_i \in \mathbb{R}^3$ represents the Principal Gaze Feature (PGF), with detailed implementation provided in Section 4.4 (2). The resulting PGFs reside on the surface of a 3D sphere, preserving the gaze-aware topology demonstrated in Fig. 2(c).

**Spherical Fitting** We adopt a Spherical Fitting (SF) algorithm to map projected features PGFs to gaze angles analytically. Following AGG (Bao & Lu, 2024), we estimate the sphere center $O_c$ and apply a rotation $R$ to align the coordinate system: $\mathbf{p}_i' = R(\mathbf{p}_i - O_c) = (x_i', y_i', z_i')^\top$. Euler angles $(\theta_i', \psi_i')$ are then derived from the rotated point $p_i'$ via physically interpretable projections:

$$\theta_i' = k_1 \arctan\left(\frac{x_i'}{z_i'}\right) + b_1, \ \psi_i' = k_2 \arcsin(y_i') + b_2, \tag{1}$$

where $(k_1, k_2, b_1, b_2)$ are learnable scaling and biases parameters. The final 3D gaze vector is: $y_i' = \text{SF}_{\theta_s}(p_i)$, $\theta_s = \{O_c, R, k_1, k_2, b_1, b_2\}$. Parameters $\theta_s$ are optimized by minimizing angular error:

$$\min_{\theta_s} \sum_{i=1}^{N} \mathcal{L}_{\text{angular}}\left(\mathbf{y}_i, \text{SF}_{\theta_s}(\mathbf{p}_i)\right). \tag{2}$$

The AGE branch ensures that small displacements in feature space correspond to proportional changes in gaze direction. Unlike AGG (Bao & Lu, 2024) which uses Isomap for final regression layer replacement, our AGE creates an intermediate geometric consistency latent space where the spherical features serve as geometric prior for adversarial alignment with the MGR branch.

Figure 4: Structure-aware modeling and view-consistent regularization. (a) The parametric eyeball is reconstructed in camera coordinate system using the predicted parameters, demonstrating structural consistency. (b) Projection of the iris and pupil regions onto 2D plane, yielding the corresponding semantic masks through weak supervision of sparse edge points. (c) VCR: Enforcing consistency in gaze prediction space and structural edge projections under simulated 3D rotations.

### 3.3 MODEL-GUIDED RECONSTRUCTION

While AGE enforces geometric alignment, it ignores anatomical constraints. MGR branch addresses this by introducing a parametric 3D eyeball, ensuring gaze prediction is anatomically plausible.

**Parametric Eyeball Modeling** As shown in Fig. 4(a). Consistent with common practice in anatomical eye modeling (Świrski & Dodgson, 2013; Dierkes et al., 2018; Popovic et al., 2023; Xiao et al., 2025), for each input frame $I$, the eyeball is represented as a sphere with parameters $\mathbf{E}_{para} = \{\mathbf{r_e}, \mathbf{r_i}, \mathbf{T}, \mathbf{r_p}, \mathbf{R}\}$, where $r_e, r_i, r_p$ denote eyeball, iris and pupil radii. $T \in \mathbb{R}^3$ indicates the eyeball center in the camera coordinate system, and $\mathbf{R_n} \in SO(3)$ describes its orientation. The optical axis is estimated as $g = \frac{o_i - o_e}{\|o_i - o_e\|}$, with $o_e$ and $o_i$ denoting the eyeball and iris center. Camera intrinsics are jointly estimated under a pinhole assumption with $f_x = f_y = f$ and $(c_x, c_y) = (W/2, H/2)$. While anatomical deviations exist (e.g., corneal protrusion), the dominant spherical geometry, combined with limited pupil movement ranges, provides a strong structural prior.

**Deformation and Projection** Following De$^2$Gaze (Xiao et al., 2025), pupil and iris point clouds are generated in the canonical space as concentric disks and rings:

$$P_p^C = \{(r_p \rho \cos(\theta), r_p \rho \sin(\theta), -L_p) \mid \rho \in [0,1], \theta \in [0,2\pi]\},$$
$$P_i^C = \{(r\cos(\theta), r\sin(\theta), -L_p) \mid r = r_p + \rho(r_i - r_p), \rho \in [0,1], \theta \in [0,2\pi]\}. \quad (3)$$

These 3D points are transformed to camera space via $[R|T]$ and projected with intrinsic $K$ onto the 2D image: To render the semantic supervision signals, we transform the canonical point clouds into the camera coordinate system by applying the predicted rotation $R$ and translation $T$: $P_p^{2D} = K[R|T]P_p^C$, $P_i^{2D} = K[R|T]P_i^C$. As shown in Fig. 4(b), only iris/pupil edges are supervised to reduce cost. Unlike De$^2$Gaze (Xiao et al., 2025) which focuses on decoupled representation for temporal tracking, our MGR branch enforces physically grounded predictions, interpretable feedback, and improved robustness to sparse supervision and domain shifts. And the AGE branch can implicitly compensate for systematic biases not captured by the MGR's physical model.

### 3.4 VIEW-CONSISTENT REGULARIZATION

Unlike 3DGazeNet (Ververas et al., 2024) which enforces multi-view consistency with explicit multi-view data, as shown in Fig. 4(c), we introduce View-Consistent Regularization (VCR) to achieve rotation-equivariant consistency without multi-view input. VCR learns a feature-space rotation operator that captures the underlying physical transformation, enabling robust generalization to unseen viewpoints. By combining MGR and AGE with adversarial learning, VCR extends consistency constraints from pure geometry to encompass both structural and feature-level representations.

**Feature Rotation** Let $f_i \in \mathbb{R}^d$ be the encoder feature. A known rotation $P \in SO(3)$ in the gaze space is mapped to the feature space via a learned linear operator $W \in \mathbb{R}^{d \times 3}$: $f_i' = f_i T_p$, $T_p = WPW^\dagger$, where $W^\dagger$ denotes the pseudo-inverse of $W$, ensuring that the feature-space rotation $T_p$ approximates the geometric transformation of the gaze-space rotation $P$.

**Geometric Prediction Consistency.** Given the original features $f_i$ and rotated features $T_p(f_i)$, the decoder $D$ produces corresponding gaze predictions: $g_i = D(f_i)$ and $\mathbf{g}_i' = D(T_p(f_i))$. We enforce consistency through: $\mathbf{g}_i' \approx P\mathbf{g}_i$, ensuring that rotating the input features produces gaze predictions consistent with applying the physical rotation directly.

**Structural Projection Consistency**  From MGR, the canonical 3D point clouds of the pupil/iris $(P_p^{3D}, P_i^{3D})$ are rotated by the same rotation $P$:

$$\widetilde{P}_p^{3D} = PP_p^{3D}, \; \widetilde{P}_i^{3D} = PP_i^{3D}. \tag{4}$$

Both original and rotated point clouds are projected with camera intrinsics $K$ for 2D supervision:

$$P_p^{2D} = KP_p^{3D}, \; \widetilde{P}_p^{2D} = K\widetilde{P}_p^{3D}, \; P_i^{2D} = KP_i^{3D}, \; \widetilde{P}_i^{2D} = K\widetilde{P}_i^{3D}. \tag{5}$$

By jointly constraining feature transformations, gaze predictions, and structural projections, VCR enforces comprehensive rotation-equivariance across the entire representation pipeline, significantly enhancing cross-view generalization while maintaining physical plausibility.

### 3.5 Loss Functions

We design a unified optimization framework that jointly trains the AGE and MGR branches, enhanced by adversarial alignment and View-Consistent Regularization.

**AGE Branch Loss**  The AGE branch is supervised by the angular error between predicted and ground-truth gaze directions. For $N$ samples with weight $w_{\text{AGE}}$, the loss is defined as:

$$L_{AGE} = w_{\text{AGE}} \frac{1}{N} \sum_{n=1}^{N} \arccos \left( \frac{\hat{\mathbf{g}}_{AGE}^n \cdot \mathbf{g}_{GT}^n}{\|\hat{\mathbf{g}}_{AGE}^n\| \|\mathbf{g}_{GT}^n\|} \right). \tag{6}$$

**MGR Branch Loss**  The MGR branch reconstructs the 3D eyeball and is supervised through both structural projection and gaze consistency. For $N$ frames and $K$ edge points with weights $w_1$ and $w_2$:

$$L_{MGR}^{edge} = w_1 \frac{1}{NK} \sum_{n=1}^{N} \sum_{k=1}^{K} \|P_{nk}^{2D} - P_{n\arg\min_j \|P_{nk}^{2D} - P_{nj}^{\text{GT,2D}}\|}^{\text{GT,2D}}\|, \; L_{MGR}^{gaze} = w_2 \frac{1}{N} \sum_{n=1}^{N} \arccos \left( \frac{\hat{\mathbf{g}}^n \cdot \mathbf{g}^n}{\|\hat{\mathbf{g}}^n\| \|\mathbf{g}^n\|} \right). \tag{7}$$

**Adversarial Alignment Loss**  For the two branches features $f_{\text{AGE}}$ and $f_{\text{MGR}}$, the feature alignment between AGE and MGR branches is enforced through a min-max game with discriminator $D$:

$$L_{\text{adv}} = E[\log D(f_{\text{AGE}})] + E[\log(1 - D(f_{\text{MGR}}))], \tag{8}$$

**VCR Loss**  Following Sec 3.4, VCR enforces two view-consistency constraints to ensure robustness under viewpoint changes. $P \in \mathbb{R}^{3 \times 3}$ is the ground-truth rotation matrix applied to $\mathbf{g}_i$.

$$L_{VCR}^{gaze} = w_{\text{gaze}}' \frac{1}{N} \sum_{i=1}^{N} \|\mathbf{g}_i' - P\mathbf{g}_i\|_2^2, \quad L_{VCR}^{edge} = w_{\text{proj}}' \frac{1}{N} \sum_{i=1}^{N} \left( \|P_p^{2D} - \widetilde{P}_p^{2D}\|_2^2 + \|P_i^{2D} - \widetilde{P}_i^{2D}\|_2^2 \right). \tag{9}$$

## 4 Experiments

### 4.1 Datasets and Preprocessing

**Dataset**  We evaluate on two real-world datasets and one synthetic dataset: **(1) TEyeD**(Fuhl et al., 2021). Comprising 20M+ images from 132 subjects across diverse scenarios, with 2D/3D landmarks, segmentation, gaze vectors, and eye movement labels. We adopt three subject-disjoint splits: $D_{T_1}$ (200k train / 30k test, 16 subjects), $D_{T_2}$ (200k train / 30k test, 16 subjects), and $D_S$ (50k train / 12k test, 39 subjects). These subsets support within-subset and cross-subset performance validation. **(2) LPW** (Tonsen et al., 2016). Collected under daily-life conditions with large illumination, pose, and occlusion variations. $D_L$: We sample 97k train / 27k test images from 22 subjects, excluding blurred or occluded frames. This dataset serves as a target domain for cross-dataset evaluation under weak 2D edge supervision. **(3) UnityEyes** (Wood et al., 2016). A synthetic dataset with accurate 3D labels and structure annotations. We use 200k images (160k train / 40k test) from 20 virtual subjects, mainly for backbone pretraining, spherical feature validation, and enforcing rotation-consistency, complementing limited real-world labels. All datasets are temporally downsampled from 25 Hz to 6.25 Hz to ensure significant eye movements and avoid redundant frames. **Implementation Details** Input eye images are cropped to $320 \times 240$ and normalized. The training dataset is augmented augmented with Gaussian ambiguity (std 1.0–2.0), random noise (0–30%), and horizontal flip (20% probability). Training is performed with batch size 128, each containing 4 consecutive frames. In the MGR branch, structural templates provide weak supervision of iris and pupil boundaries. Specifically, 26 iris edge points are sampled by uniformly distributing angles $\theta$ over $\theta \in [0, 0.1\pi] \cup [0.9\pi, 1.1\pi] \cup [1.9\pi, 2\pi]$ with radius $\rho \in [0, 1]$, while 128 pupil contour points are evenly sampled over $[0, 2\pi]$. These serve as pseudo-labels for 2D structural projection. For Isomap, we adopt the Scikit-learn implementation with 300 neighbors in geodesic distance. Optimization uses LAMB (You et al., 2019) with initial learning rate $2 \times 10^{-3}$ and weight decay 0.02.

Table 1: Comparison of SG-Gaze with baseline models under cross-dataset and within-dataset evaluation settings. 3D Gaze error is in degrees (lower is better, ▼ denotes reduction, ▲ denotes increase).

| Method | $D_{T_1} \to D_S$ | $D_{T_1} \to D_L$ | $D_{T_1} \to D_{T_2}$ | $D_{T_2} \to D_S$ | $D_{T_2} \to D_L$ | $D_{T_2} \to D_{T_1}$ | within $D_{T_1}^*$ | within $D_{T_2}^*$ |
|---|---|---|---|---|---|---|---|---|
| ResNet-18 | 5.02 | 6.83 | 3.20 | 5.31 | 6.68 | 3.29 | 1.68 | 1.59 |
| ResNet-18+SG-Gaze | 4.10▼18.32% | 5.07▼25.77% | 2.07▼35.31% | 3.93▼25.99% | 4.87▼27.10% | 2.03▼32.22% | 1.11▼33.93% | 1.33▼16.35% |
| ResNet-50 | 4.04 | 5.47 | 2.54 | 5.16 | 5.14 | 3.16 | 1.32 | 1.34 |
| ResNet-50+SG-Gaze | 3.91▼3.21% | 3.75▼31.44% | 1.88▼25.78% | 4.36▼15.50% | 4.21▼18.09% | 1.94▼38.61% | 1.22▼9.10% | 1.09▼18.66% |
| VGG-16 | 5.5 | 7.14 | 3.61 | 5.94 | 7.61 | 3.94 | 2.12 | 2.35 |
| VGG-16+SG-Gaze | 5.13▼6.73% | 5.2▼27.02% | 3.30▼8.59% | 5.27▼11.28% | 5.33▼29.97% | 2.97▼24.62% | 2.78▲31.13% | 3.13▲33.19% |

Table 2: Cross-domain comparison with state-of-the-art gaze estimation methods. 3D Gaze error is in degrees (lower is better). SG-Gaze achieves the best generalization across unseen domains, benefiting from its structurally and geometrically consistent representation.

| Method | $D_{T_1}\to D_S$ | $D_{T_1}\to D_{T_2}$ | $D_{T_1}\to D_L$ | $D_{T_2}\to D_{T_1}$ | $D_{T_2}\to D_S$ | $D_{T_2}\to D_L$ | $D_S\to D_{T_1}$ | $D_S\to D_{T_2}$ | $D_S\to D_L$ | $D_L\to D_{T_1}$ | $D_L\to D_{T_2}$ | $D_L\to D_S$ |
|---|---|---|---|---|---|---|---|---|---|---|---|---|
| RAT (Bao et al., 2022) | 5.82 | 2.83 | 5.73 | 3.91 | 5.31 | 5.97 | 6.18 | 5.29 | 4.79 | 5.68 | 4.59 | 4.83 |
| Latentgaze (Lee et al., 2022) | 5.5 | 3.14 | 5.81 | 3.61 | 5.94 | 5.83 | 5.11 | 4.99 | 4.83 | 5.12 | 4.35 | 5.02 |
| FFGaze (Zhang et al., 2017) | 5.04 | 3.47 | 5.44 | 3.54 | 5.16 | 5.44 | 4.96 | 5.16 | 5.24 | 5.32 | 4.34 | 4.77 |
| PureGaze (Cheng et al., 2022) | 4.5 | 2.94 | 4.90 | 2.61 | 4.94 | 5.71 | 4.61 | 5.94 | 5.21 | 5.03 | 4.35 | 4.97 |
| AGG (Bao & Lu, 2024) | 4.0 | 2.21 | 4.33 | 2.15 | 4.21 | 4.94 | 3.44 | 4.53 | 4.21 | 4.02 | 3.58 | 4.44 |
| De$^2$Gaze (Xiao et al., 2025) | 4.9 | 2.88 | 5.10 | 3.22 | 5.02 | 5.88 | 4.79 | 5.25 | 4.97 | 5.10 | 4.72 | 5.33 |
| Baseline | 5.02 | 3.20 | 6.83 | 3.29 | 5.31 | 6.68 | 5.21 | 5.92 | 5.41 | 5.32 | 4.77 | 5.63 |
| ResNet18+SG-Gaze | 4.10 | 2.07 | 5.07 | 2.03 | 3.93 | 4.87 | 3.41 | 4.36 | 4.43 | 4.12 | 3.24 | 4.23 |
| ResNet50+SG-Gaze | 3.91 | 1.88 | 3.75 | 1.94 | 4.36 | 4.21 | 3.29 | 4.04 | 4.18 | 3.87 | 3.35 | 4.01 |

## 4.2 MAIN RESULTS

**Cross-dataset Evaluation** We evaluate SG-Gaze by integrating it into different backbones in both domains and within the domain (Table 1). For ResNet-18, SG-Gaze reduces gaze error by up to **35.31%** in $D_{T_1} \to D_{T_2}$ transfer and over **25%** on most tasks. ResNet-50+SG-Gaze achieves **26.77%–38.61%** gains. SG-Gaze intentionally introduces stronger physical inductive biases for structural and cross-view consistency, which may slightly reduce fine-grained fitting in low-capacity models (e.g., VGG-16) but delivers substantial cross-domain improvements. This favorable trade-off aligns with real-world requirements where domain shift is inevitable, demonstrating gaze estimation generalization, independent of backbone architecture or domain shift.

**Comparison with SOTA Methods** We further benchmark SG-Gaze against recent state-of-the-art generalizable approaches, including RAT (Bao et al., 2022), LatentGaze (Lee et al., 2022), FFGaze (Zhang et al., 2017), De$^2$Gaze (Xiao et al., 2025), AGG( (Bao & Lu, 2024)) and PureGaze (Cheng et al., 2022). As shown in Table 2, SG-Gaze achieves the best or second-best accuracy in nearly all 12 cross-domain transfer tasks. For example, ResNet-50+SG-Gaze reduces the error to **1.88°** on $D_{T_1}\to D_{T_2}$ and **1.94°** on $D_{T_2}\to D_{T_1}$, outperforming the strongest baseline (AGG) by an average of **7.7%**. SG-Gaze also shows clear advantages in challenging scene transfers such as $D_S\to D_{T_1}$ and $D_S\to D_{T_2}$. These results highlight that SG-Gaze not only enhances backbone models but also surpasses existing SOTA methods, demonstrating generalization across diverse domains.

**Within-dataset Evaluation** We further evaluate SG-Gaze within-domain. As shown in Table 4, SG-Gaze achieves competitive within-domain performance (0.94° with ResNet18), while maintaining high efficiency (11.68M parameters, 11.24G FLOPs). Notably, it also improves semantic edge IoU and 2D eyeball center localization, validating enhanced geometric and structural consistency. Our work prioritizes cross-domain generalization and efficiency over within-domain leaderboard performance, addressing the critical challenge of domain shift in real-world applications.

## 4.3 ABLATION STUDIES

We perform ablations on two TEyeD subsets ($D_{T_1}$, $D_{T_2}$) to evaluate the contribution of each component by selectively removing or replacing modules. The results are summarized in Table 3.

**(1) The Effect of Analytical Gaze Estimation** Removing the AGE branch eliminates physically interpretable analytical supervision ($w/o$ AGE), leading to a notable error increase to **2.57°** on $D_{T_1} \to D_{T_2}$. Although semantic and center-based constraints still offer auxiliary guidance, the model loses explicit geometric supervision that aligns predicted gaze with analytical eye geometry. This confirms the necessity of integrating analytical geometric priors with data-driven learning.

**(2) The Effect of Model-Guided Reconstruction** Removing the MGR branch ($w/o$ MGR) leads to the most severe degradation, since no 3D edge projection loss is applied and the structural cues from iris/pupil edges are absent. For instance, in the cross-domain setting $D_{T_1} \to D_{T_2}$, the 3D gaze

Table 3: Ablation study on two TEyeD subsets. We evaluate the complete SG-Gaze model against variants with AGE, MGR, or VCR removed. The full model achieves the best performance across all metrics, demonstrating that each component contributes positively to accuracy and robustness.

| Settings | Loss | $D_{T_1} \to D_{T_2}$ | | | | | within $D_{T_1}^*$ | | | | |
|---|---|---|---|---|---|---|---|---|---|---|---|
| | | 3D gaze [°]↓ | 2D gaze [°]↓ | Sem. Iou | 2D pupil cent.[px]↓ | 2D eye cent.[px]↓ | 3D gaze [°]↓ | 2D gaze [°]↓ | Sem. Iou | 2D pupil cent.[px]↓ | 2D eye cent.[px]↓ |
| w/o MGR Module | Gaze | 3.11 | 10.88 | N/A | N/A | N/A | 2.86 | 11.75 | N/A | N/A | N/A |
| w/o AGE Module | Gaze | 2.57 | 9.76 | N/A | N/A | N/A | 1.83 | 10.32 | N/A | N/A | N/A |
| | Sem. + Gaze + Cent. | 2.72 | 11.31 | 86.9% | 5.12 | 9.38 | 2.54 | 9.72 | 87.9% | 1.96 | 2.65 |
| w/o VCR | Gaze | 2.07 | 9.36 | N/A | N/A | N/A | 1.53 | 8.12 | N/A | N/A | N/A |
| | Sem. + Gaze + Cent. | 2.69 | 10.33 | 87.9% | 4.29 | 8.27 | 2.04 | 9.22 | 88.1% | 1.63 | 2.27 |
| SG-Gaze (**Ours**) | Gaze | **1.88** | **8.02** | N/A | N/A | N/A | **0.94** | **6.22** | N/A | N/A | N/A |
| | Sem. + Gaze + Cent. | 2.41 | 9.24 | **88.6%** | **3.44** | **8.12** | 1.11 | 8.82 | **92.1%** | **0.74** | **1.98** |

| Methods | Backbone | Params /(M) | FLOPs /(G) | Loss | TEyeD-$D_{T_1}$ | | | |
|---|---|---|---|---|---|---|---|---|
| | | | | | 3D gaze [°]↓ | 2D gaze [°]↓ | Sem. Iou | 2D eye cent.[px]↓ |
| (Fuhl et al., 2021) | ResNet50 | 26.08 | 24.97 | Gaze | 1.88 | 6.90 | N/A | N/A |
| (Kim et al., 2019) | CNN | 0.16 | 0.14 | Gaze | 3.65 | 8.34 | N/A | N/A |
| Transformer-based | ResNet18 | 15.53 | 14.83 | Gaze | 1.57 | 6.36 | N/A | N/A |
| (Vaswani, 2017) | ResNet50 | 27.66 | 24.98 | Gaze | 1.77 | 6.40 | N/A | N/A |
| QueryDETR | ResNet18 | 16.36 | 18.33 | Gaze | 3.12 | 8.01 | N/A | N/A |
| (Carion et al., 2020) | ResNet50 | 29.89 | 28.42 | Gaze | 3.08 | 7.86 | N/A | N/A |
| Nikola et al. | ResNet50 | 28.56 | 26.44 | Gaze | 1.04 | 7.40 | N/A | N/A |
| | ResNet50 | 28.56 | 26.44 | Sem. | 20.16 | 39.10 | 92.5% | 11.41 |
| (Popovic et al., 2023) | ResNet50 | 28.56 | 26.44 | S+G+C | 1.21 | 10.39 | 91.4% | 2.02 |
| De²Gaze | ResNet18 | 14.48 | 13.56 | Gaze | **0.54** | **5.43** | N/A | N/A |
| | ResNet18 | 14.48 | 13.56 | Sem. | 20.12 | 39.09 | **94.2%** | 11.41 |
| (Xiao et al., 2025) | ResNet18 | 14.48 | 13.56 | S+G+C | 0.96 | 7.6 | 93.4% | **1.52** |
| SG-Gaze (Ours) | ResNet18 | **11.68** | **11.24** | Gaze | 0.94 | 6.22 | N/A | N/A |
| | ResNet18 | 11.68 | 11.24 | Sem. | 21.12 | 39.69 | 93.3% | 12.22 |
| | ResNet18 | 11.68 | 11.24 | S+G+C | 1.11 | 8.82 | 92.1% | 1.88 |

Table 4: The table reports backbone, Params, FLOPs and loss settings. SG-Gaze achieves the best trade-off between accuracy, efficiency and semantic consistency without domain shift.

Figure 5: Ablation study heatmap showing the impact of removing different modules (AGE, MGR, VCR) on gaze and semantic metrics, Darker color indicates better performance.

| Metrics \ Settings | w/o MGR | w/o AGE | w/o VCR | Ours |
|---|---|---|---|---|
| 3D Gaze (T1→T2) | 3.11 | 2.72 | 2.69 | 2.41 |
| 2D Gaze (T1→T2) | 10.88 | 11.31 | 10.33 | 9.24 |
| IoU (T1→T2) | | 86.90 | 87.90 | 88.60 |
| 2D Pupil (T1→T2) | | 5.12 | 4.29 | 3.44 |
| 2D Eye (T1→T2) | | 9.38 | 8.27 | 8.12 |
| 3D Gaze (within) | 2.06 | 1.83 | 2.04 | 1.82 |
| 2D Gaze (within) | 11.75 | 9.72 | 8.22 | 6.38 |
| IoU (within) | | 87.90 | 88.10 | 92.10 |
| 2D Pupil (within) | | 1.96 | 1.63 | 0.74 |
| 2D Eye (within) | | 2.65 | 2.27 | 1.98 |

error increases from **1.88°** to **3.11°**. This confirms that enforcing edge-based structural constraints is crucial for recovering geometric consistency and achieving robust gaze estimation.

**(3) The Effect of View-Consistent Regularization** Discarding the VCR module (*w/o* VCR) weakens robustness against cross-view perturbations. The 3D gaze error rises from **1.88°** to **2.07°** in cross-domain evaluation, showing that rotation-consistency regularization not only augments the training distribution with view perturbations but also enhances cross-view generalization.

**(4) Ablation Heatmap** As shown in Fig. 5, the complete model consistently outperforms all ablated versions across metrics. These results highlight that multi-task joint training is essential for balanced performance. Furthermore, semantic IoU and 2D localization achieve the best results, confirming that our unified framework improves both geometric fidelity and physical consistency.

### 4.4 Efficient Learning under Limited Supervision

**(1) Few-Shot Adaptation** Accurate gaze estimation is limited by the scarcity of reliable 3D gaze annotations in head-mounted datasets (Kim et al., 2019; Fuhl et al., 2021), while iris and pupil masks are abundant and easier to label. To leverage this imbalance, we adopt a two-stage scheme: pre-train on large-scale semantic labels, then fine-tune with few 3D gaze samples. As shown in Fig. 6, training from scratch with limited annotations performs poorly, whereas semantic pre-training provides a reasonable zero-shot baseline. Finally, even with only 1% labeled gaze samples per subject, combining pre-training with light fine-tuning yields the largest improvement.

**(2) Isomap Selection and Isometric Propagator** We evaluate three dimensionality reduction methods to project features $f_i$ into 3D space. As shown in Fig. 7, gaze labels form a smooth spherical manifold. Isomap best preserves the global spherical geometry of gaze, while LLE (Roweis & Saul, 2000) collapses global structure and t-SNE (Maaten & Hinton, 2008) distorts geometry and amplifies noise. To enable differentiable training, we introduce a lightweight Isometric Propagator **(IP)**—a 3-layer MLP trained to approximate Isomap embeddings with high fidelity (L1 error 0.05, angular error $< 0.3°$), then frozen during main training. This design maintains geometric fidelity while avoiding Isomap's computational cost, making it ideal for gaze representation learning.

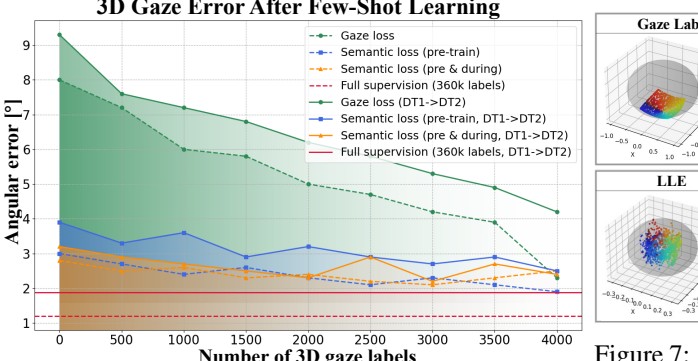

Figure 6: Few-shot fine-tuning results showing that semantic pre-training enables consistent performance gains within and across datasets, even with very limited gaze annotations.

Figure 7: 3D visualization of gaze representations on UnityEyes after dimensionality reduction (Isomap, t-SNE, and LLE). only Isomap preserves the global manifold and aligns well with GT gaze directions.

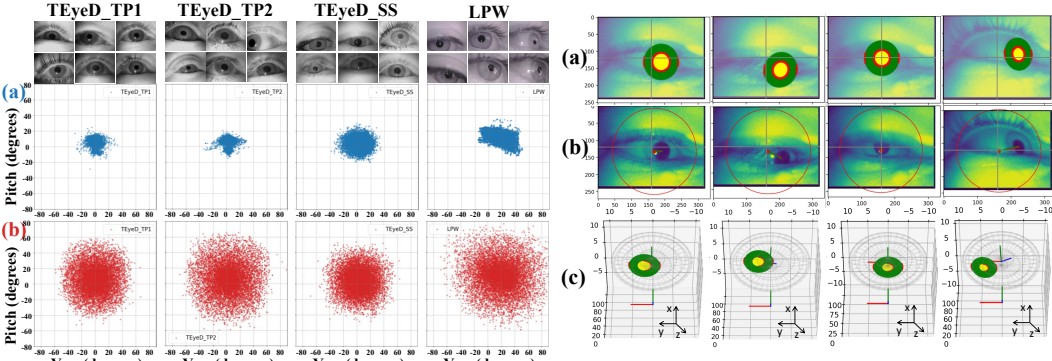

Figure 8: Distribution of gaze. (a) Without VCR, predictions concentrate within a narrow range. (b) With VCR, distributions become more diverse and evenly spread, covering a larger field-of-view.

Figure 9: (a) 2D projection of pupil/iris from MGR branch (red areas represent the edge sampling points). (b) Predicted gaze (green) vs. ground truth (red). (c) Reconstructed 3D eyeball model in camera space.

## 4.5 QUALITATIVE RESULTS

**(1) VCR-Enhanced Gaze Distribution Diversity** As shown in Fig. 8(a), models without VCR produce clustered predictions biased toward dominant training viewpoints, severely limiting gaze coverage. In contrast, VCR enables a more diverse and uniform distribution across extreme angles [Fig. 8(b)], effectively bridging the gap between controlled and in-the-wild domains. **(2) VCR-Enhanced Rotation Equivariance** We provide direct physical validation by comparing ground-truth distributions under physical 3D rotations $P$ against SGR transformed by $T_P$ and projected via Isomap (Fig. 10). High consistency between the two distribution visualizations and quantitative results (gaze equivariance error $\varepsilon_{\text{gaze}} < 0.96°$, structural consistency error $\varepsilon_{\text{struct}} < 1.88\text{px}$) confirm that our feature-space rotation $T_P$ effectively approximates genuine 3D physical transformations $P$.

**(3) Prediction and Rendering Visualization** The 3D eye model and estimated gaze direction are projected onto the 2D image plane. As shown in Fig. 9, the predicted 3D gaze vectors [Fig. 9(b)] remain consistent with the eye's viewing direction across frames, confirming estimation reliability. The reconstructed 3D eye model [Fig. 9(c)] is well aligned with pupil and iris projections [Fig. 9(a)], demonstrating accurate structural recovery and geometrically plausible 2D rendering.

## 4.6 DISCUSSION

**(1) Robustness to Low-Quality or Noisy Inputs** Although SG-Gaze is trained on near-eye images, it does not rely on high-resolution textures. AGE focuses on geometric structure, MGR uses sparse edge cues, and VCR introduces blur-, scale-, and view-perturbations, collectively enabling robustness under low-resolution or degraded inputs. Even when edge cues become unreliable in noisy real-world conditions, the dual-branch design provides complementary fallback: AGE supplies geometric consistency, MGR contributes structural priors, and inference requires no 2D semantics.

Figure 10: Feature-space rotation validation. (a) GT distributions under 3D physical rotations $P$ around X/Y/Z axes. (b) SGR transformed by $T_P$ and projected via Isomap maintain a high degree of consistency with the GT distribution after physical rotation on all rotation axes.

Experiments on TEyeD and LPW verify that SG-Gaze maintains reliable performance despite variations in quality, illumination, and segmentation noise.

**(2) Modeling Simplifications and Trade-offs** SG-Gaze intentionally prioritizes subject-invariant geometric consistency over subject-specific anatomy (e.g., kappa angle), trading slight in-domain accuracy losses for stronger cross-domain robustness. Likewise, weak 2D edge supervision emphasizes physically plausible 3D structure rather than pixel-level 2D precision. Although MGR adopts simplified eyeball and camera models, but AGE and adversarial alignment help compensate for such model mismatches, and device-specific fine-tuning or lightweight correction terms can further adapt the model for high-precision scenarios. Consistent gains across 12 heterogeneous transfer tasks indicate that SG-Gaze benefits from learned geometric invariances rather than dataset-specific bias.

## 5 RELATED WORK

**Model-based Methods** Model-based approaches reconstruct eyeball's anatomical structure (Chen et al., 2008; Hennessey et al., 2006; Wood & Bulling, 2014) using infrared glints, corneal reflections (Hansen & Ji, 2009), or parametric 3D models (Smith et al., 2020; Popovic et al., 2023). Differentiable rendering (Wood et al., 2019; Yoon et al., 2021), neural implicit surfaces (Guo et al., 2022), and hybrid pipelines (Zhang et al., 2021) have improved geometric fidelity and robustness. Recent works extend model-based estimation with representation learning and adaptability: De$^2$Gaze (Xiao et al., 2025) proposed a 3D eye tracking method using deformable and decoupled representations with lightweight time-series. Few-shot personalization (Wang et al., 2022a) and uncertainty-aware fitting (Kim et al., 2021) further enhance cross-subject adaptability.

**Appearance-based Methods** Deep learning has driven most recent progress in gaze estimation, leveraging large-scale datasets (Krafka et al., 2016; Zhang et al., 2020; 2015; Kellnhofer et al., 2019; Zhang et al., 2017). Models predictions from eye crops (Zhang et al., 2015; Cheng et al., 2020b), full-face images (Bao et al., 2021; Balim et al., 2023; Chen & Shi, 2018; Cheng et al., 2020a; Krafka et al., 2016; Zhang et al., 2017), or their fusion (Krafka et al., 2016; Park et al., 2020). Leveraging two-eye asymmetry has been shown to improve prediction accuracy (Chen & Shi, 2018; Cheng et al., 2020b). Recent approaches employ attention mechanisms, hierarchical and multi-stream architectures, as well as transformer-based backbones (Qin et al., 2025; Yin et al., 2024; Miyato et al., 2023). **Cross-domain Gaze Estimation** Existing methods include unsupervised domain adaptation (Lahiri et al., 2018; Zhang et al., 2022a), contrastive objectives (Wang et al., 2022b) and collaborative learning with outlier guidance (Liu et al., 2021). But these methods require target-domain data. Techniques include rotation-consistent regularization (Bao et al., 2022), source-domain feature purification (Cheng et al., 2022), style perturbation (Zhou et al., 2024), and synthetic imagery (Zhang et al., 2022b). Recent work incorporate geometric constraints and 3D structure. 3D eye mesh regression (Ververas et al., 2024) and geometry projections with spherical training (Bao & Lu, 2024) enhance cross-domain robustness without target-domain access.

## 6 CONCLUSION

In this work, we proposed SG-Gaze, a physically guided dual-branch framework that integrates analytical gaze estimation, model-guided reconstruction, and view-consistent regularization. Our approach enforces structural and geometric consistency while promoting rotation-equivariant feature learning, yielding accurate and generalizable 3D gaze representations. Extensive experiments demonstrate its strong cross-domain generalization and improved gaze diversity, highlighting the benefits of incorporating physical priors into learned representations. SG-Gaze provides a foundation for scalable, few-shot, and personalized gaze estimation. Future work will explore learning temporal and personalized gaze representations from larger in-the-wild datasets, aiming to improve cross-subject generalization and adaptation in real-world AR/VR environments.

## ETHICS STATEMENT

All authors have read and agree to abide by the ICLR Code of Ethics[1]. This work focuses on 3D gaze estimation and reconstruction using eye images from publicly available datasets. No experiments involve human or animal subjects directly, and no personally identifiable or private information is collected. All datasets used (e.g., TEyeD (Fuhl et al., 2021), LPW (Tonsen et al., 2016), UnityEyes (Wood et al., 2016)) are widely adopted in the research community and comply with their respective licenses. To further ensure ethical compliance, we do not release any raw personal eye-tracking data, but only model code and trained weights for reproducibility.

The potential benefits of this research include applications in AR/VR interaction, assistive technologies for people with motor impairments, and clinical support tools (e.g., vision diagnostics). However, we acknowledge that gaze tracking technologies can raise concerns regarding privacy, surveillance, and potential misuse in sensitive contexts. Our method improves accuracy and generalizability, but it should be applied responsibly and only in scenarios that respect user consent and privacy.

We encourage the research community to use our contributions solely for beneficial applications and in accordance with ethical guidelines. We commit to releasing our code, model architectures, and training details in line with the standards of reproducibility and transparency. No conflicts of interest or external sponsorships that might have influenced this work exist.

## REPRODUCIBILITY STATEMENT

We take reproducibility seriously and have taken multiple measures to ensure that all aspects of our work can be replicated by other researchers. First, we provide anonymized source code as supplementary material, which includes the complete implementation of the proposed SG-Gaze framework, detailed configuration files, and training/evaluation scripts.

Second, we explicitly report all hyperparameters such as optimizer type, learning rate, weight decay, batch size, training epochs, and initialization strategies, making it possible to reproduce our training schedules.

Third, for our proposed View-Consistent Regularization (VCR), we include the complete mathematical formulation, proof sketches, and implementation details in the Appendix to clarify all theoretical claims.

Fourth, additional ablation studies, sensitivity analyses, and visualizations are provided in the supplementary material to support the design choices and robustness of the method.

Finally, we encourage further validation by including scripts to reproduce all figures and tables reported in the paper.

Together, these efforts ensure that our results are reproducible and provide a solid foundation for future research on interpretable and generalizable gaze estimation.

## USE OF LARGE LANGUAGE MODELS (LLMS)

In line with the ICLR 2025 policy on the use of large language models (LLMs), we provide a transparent account of how such tools were utilized in the preparation of this paper. We stress that the scientific contributions, including the formulation of the problem, the design of the method, the theoretical derivations, and the experimental implementation, were entirely conceived and executed by the authors. The role of LLMs was restricted to auxiliary support as outlined below:

- **Writing and Editing:** LLMs (e.g., GPT-based models) were used to assist with grammar checking, sentence restructuring, and improving the clarity and readability of the manuscript. This included rephrasing certain technical descriptions for conciseness and consistency of style across sections.

---

[1]https://iclr.cc/public/CodeOfEthics

- **Literature Organization:** LLMs were used to generate initial summaries of related work from bibliographic entries provided by the authors. These summaries were manually reviewed, corrected, and integrated into the final Related Work section by the authors.

- **Proofreading and Consistency Checking:** LLMs assisted in detecting potential inconsistencies in notation and formatting across the paper (e.g., consistent use of symbols for loss functions, dataset names, and evaluation metrics).

Importantly, LLMs were *not* used to generate research ideas, design experiments, propose model architectures, perform data analysis, or interpret results. All conceptual insights, technical innovations, and conclusions are solely attributed to the authors. We include this statement to ensure transparency and compliance with the ICLR Code of Ethics regarding responsible use of LLMs.

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

## APPENDIX

Section A introduces our demo video information. A live demo of our method can be found at this link. Section B describes the implementation process of the method in more details. We have carried out additional experiments, and the results will be explained in Section C. Finally, in Section D, we will discuss the limitations of the method and future work. Our code will be released on GitHub upon acceptance.

## A    DEMO VIDEO

To better illustrate our proposed method, we provide a demo video as supplementary material. The video demonstrates the complete pipeline of our 3D gaze estimation framework, including raw eye image input, 2D rendering, 3D eyeball reconstruction, and mesh visualization. Specifically:

- **Original Image:** The raw infrared eye image captured by the tracker.
- **2D Rendering:** Projection of the reconstructed eyeball and pupil region onto the image plane.
- **Eye Tracker + Sphere:** Visualization of the eyeball sphere model aligned with the tracker coordinate system.

- **3D Eyeball:** Fusion of image appearance with the 3D eyeball model for interpretable gaze analysis.

- **3D Mesh:** Geometric mesh reconstruction of the eyeball with anatomical and structural priors.

The video highlights how our model integrates both appearance features and structural constraints, ensuring interpretable and physically consistent gaze estimation.

## B METHODOLOGICAL DETAILS

### B.1 RENDER SEMANTICS (MGR BRANCH)

We begin by generating the template point clouds for both the pupil and the iris using polar coordinates. The process involves discretizing the angles and radii, and then converting the polar coordinates into 3D Cartesian coordinates.

**Generate Angle Grid** We generate a set of angles in the range $[0, 2\pi)$ for each batch and frame. This is done by discretizing the angle space into $N_{\text{angles}}$ points:

$$\theta_i = \frac{2\pi i}{N_{\text{angles}}}, \quad \text{for} \quad i = 0, 1, 2, \ldots, N_{\text{angles}} - 1, \tag{10}$$

this results in an angle grid of shape $[B, N_{\text{angles}}]$, where $B$ is the batch size multiplied by the number of frames.

**Generate Radius Grids** The radius of the pupil and iris are discretized into $N_{\text{radius}}$ points. For the pupil, the radii range from 0 to $r_{\text{pupil}}$, while for the iris, they range from $r_{\text{pupil}}$ to $r_{\text{iris}}$. The radii are generated as:

$$r_{\text{pupil}}^i = \frac{i}{N_{\text{radius}}} r_{\text{pupil}},$$

$$r_{\text{iris}}^i = \frac{i}{N_{\text{radius}}}(r_{\text{iris}} - r_{\text{pupil}}) + r_{\text{pupil}}, \tag{11}$$

$$\text{for} \quad i = 0, 1, 2, \ldots, N_{\text{radius}} - 1,$$

this generates two radius grids, one for the pupil and one for the iris, both of shape $[B, N_{\text{radius}}]$.

**Convert Polar Coordinates to Cartesian Coordinates.** We convert the polar coordinates into 3D Cartesian coordinates for both the pupil and the iris. For each pair of radius $r$ and angle $\theta$, the Cartesian coordinates $(x, y, z)$ are computed as:

$$x = r \cdot \cos(\theta), \quad y = r \cdot \sin(\theta). \tag{12}$$

For the pupil and iris point clouds, the $z$-coordinate is set to a fixed value, determined by the distance from the camera, $L_p$, and inverted to place the point clouds in front of the camera:

$$z_{\text{pupil}} = z_{\text{iris}} = -L_p \tag{13}$$

Thus, the final 3D coordinates for the pupil and iris are:

$$P_{\text{pupil}} = \left(x_{\text{pupil}}, y_{\text{pupil}}, z_{\text{pupil}}\right),$$

$$P_{\text{iris}} = \left(x_{\text{iris}}, y_{\text{iris}}, z_{\text{iris}}\right). \tag{14}$$

The resulting point clouds have shapes $[B, N_{\text{angles}} \times N_{\text{radius}}, 3]$.

### B.2 AGE BRANCH DETAILS

A key objective of AGE branch is to train the feature extractor $F_{\theta_1}$ under the guidance of spherical fitting, so that gaze features are aligned with the physical definition of eye rotations. However, directly integrating the Isomap (Tenenbaum et al., 2000) algorithm into backpropagation is computationally prohibitive: the time complexity of Isomap is $O(N^2 \log N)$ and the memory complexity is

$O(N^2)$, where $N$ denotes the number of samples. Processing hundreds of thousands of gaze features with Isomap is thus infeasible in both time and memory.

**Isometric Propagator (IP)**  To overcome this limitation, we introduce the Isometric Propagator (IP) following previous studies (Bao & Lu, 2024), a lightweight three-layer MLP $IP_{\theta_3}(\cdot)$ that parameterizes the Isomap algorithm. The IP is trained to approximate Isomap embeddings during an initialization phase. Specifically, we freeze the parameters of the pretrained CNN $F_{\theta_1}$ and train $IP_{\theta_3}$ to regress the Isomap outputs from its input features:

$$\min_{\theta_3} \frac{1}{N} \sum_{i=1}^{N} L_1 \left( \text{Isomap}(f_i),\ IP_{\theta_3}(f_i) \right),$$
(15)

where $f_i = F_{\theta_1}(x_i)$ are CNN features and $\mathscr{L}_1$ denotes the L1 loss.

**Retraining the Feature Extractor**  After training, we freeze the parameters of $IP_{\theta_3}$ and replace Isomap with it for training the feature extractor. Sphere-Fitting Training objective is then defined as:

$$\min_{\theta_1} \frac{1}{N} \sum_{i=1}^{N} L_1 \left( \hat{e}_i,\ IP_{\theta_3}(F_{\theta_1}(x_i)) \right),$$
(16)

where $\hat{e}_i$ denotes the ground-truth embedding derived from spherical fitting. The Isometric Propagator is only used during source-domain training.

**Inference**  At test time, gaze is estimated through Isomap and Spherical Fitting:

$$g_i = SF_{\theta_s} \left( \text{Isomap}(F_{\theta_1}(x_i)) \right).$$
(17)

This training strategy directly optimizes gaze features according to their geometric relation with spherical fitting, ensuring physical interpretability while maintaining computational feasibility.

### B.3  TRAINING DETAILS

We use ResNet-18/50 as backbones for fair comparison, followed by our dual-branch decoder. SG-Gaze is trained on NVIDIA A100 GPU, with a batch size of 128. We set the initial weights of projection edge loss $\lambda_{edge}$, eyeball center loss $\lambda_{eye}^{center}$ and pupil center loss $\lambda_{pupil}^{center}$ to 0.15. The initial weight of that two 3D gaze loss $\lambda_{gaze}^{L2}$ and $\lambda_{gaze}^{cos-sin}$ are set to 2.5. The training process is terminated at 160 epochs. The Sphere-Fitting Training is 20 epochs, while the IP is trained for 100 epochs on 10000 randomly selected samples.

### B.4  VIEW-CONSISTENT REGULARIZATION (VCR)

**Rotation parameterization**  We use the right-handed camera coordinate system and compose the 3D viewpoint rotation as $P(\alpha, \beta, \gamma) = R_z(\gamma) R_y(\beta) R_x(\alpha) \in \text{SO}(3)$, where $\alpha, \beta, \gamma$ denote pitch (x-axis), yaw (y-axis) and roll (z-axis), respectively. Unless otherwise stated, angles are in degrees.

**Mixture-of-ranges sampling**  To simultaneously cover near-view perturbations and far-view shifts while keeping training stable, we sample $(\alpha, \beta, \gamma)$ from a two-component mixture:

$$(\alpha, \beta, \gamma) \sim (1 - \lambda)\, \Omega_{\text{loc}}\ +\ \lambda\, \Omega_{\text{glob}},$$

with $\lambda = 0.2$. The local component focuses on small perturbations

$$\Omega_{\text{loc}} = U([-12°, 12°]) \times U([-12°, 12°]) \times U([-6°, 6°]),$$
(18)

and the global component sweeps a wider FoV, dataset-aware:

$$\Omega_{\text{glob}} = U([-A_{\max}, A_{\max}]) \times \\ U([-B_{\max}, B_{\max}]) \times U([-G_{\max}, G_{\max}]).$$
(19)

For UnityEyes pretraining we use $(A_{\max}, B_{\max}, G_{\max}) = (60°, 50°, 15°)$; for TEyeD/LPW fine-tuning we use $(35°, 30°, 10°)$. This setting matches the broader synthetic coverage while avoiding excessive roll in head-mounted real captures. We further adopt a two-stage curriculum: for the first $E=10$ epochs, $\lambda=0$ (local-only), then switch to the above mixture.

**AGE branch under rotation (feature-space).** Let $f_i \in \mathbb{R}^d$ be the shared-encoder feature of frame $I_i$. We map the 3D rotation into feature space via the learned linear operator $W \in \mathbb{R}^{d \times 3}$ and its pseudo-inverse $W^\dagger$:

$$f_i' = f_i P_f, \qquad P_f = W P W^\dagger.$$

The analytical decoder $\Phi_{\text{dec}}$ (shared for original/rotated features) predicts

$$g_i = \Phi_{\text{dec}}(f_i), \qquad g_i' = \Phi_{\text{dec}}(f_i'),$$

and VCR enforces rotation-equivariant gaze prediction

$$L_{\text{VCR}}^{\text{gaze}} = w_{\text{gaze}}' \frac{1}{N} \sum_{i=1}^{N} \left\| g_i' - P g_i \right\|_2^2.$$

We do *not* re-encode images after rotation; gradients flow through $W$, $W^\dagger$ and $\Phi_{\text{dec}}$.

**MGR branch under rotation (structure-space).** From the MGR branch we have canonical pupil/iris point clouds $\text{PC}_p, \text{PC}_i$ and their camera-space instances $P_p^{3D}, P_i^{3D}$ obtained by the predicted pose $[R|T]$ and then projected by $K$ to $P_p^{2D}, P_i^{2D}$ (see main text). VCR applies the *same* geometric rotation $P$ to the camera-space point clouds (no re-run of MGR):

$$\widetilde{P}_p^{3D} = P P_p^{3D}, \quad \widetilde{P}_i^{3D} = P P_i^{3D}, \quad \widetilde{P}_p^{2D} = K \widetilde{P}_p^{3D}, \quad \widetilde{P}_i^{2D} = K \widetilde{P}_i^{3D}.$$

We enforce 2D semantic edge consistency via nearest-neighbor matching:

$$L_{\text{VCR}}^{\text{edge}} = w_{\text{proj}}' \frac{1}{N} \sum_{i=1}^{N} \left( \left\| P_p^{2D} - \widetilde{P}_p^{2D} \right\|_2^2 + \left\| P_i^{2D} - \widetilde{P}_i^{2D} \right\|_2^2 \right).$$

The total VCR regularization is

$$L_{\text{VCR}} = L_{\text{VCR}}^{\text{gaze}} + L_{\text{VCR}}^{\text{edge}},$$

which used alongside AGE/MGR losses in the joint objective. In practice we share the decoder across original/rotated features (AGE) and reuse the predicted structure for geometric rotation (MGR), which avoids extra backbone passes while enforcing consistent physics across views.

# C   ADDITIONAL EXPERIMENTS

## C.1   EFFECT OF ROTATION ANGLES IN VCR

**Motivation** Our View-Consistent Regularization (VCR) applies synthetic viewpoint perturbations during training to enforce rotation-equivariant consistency. The choice of rotation ranges may affect the trade-off between local robustness and global generalization. Here we study the influence of different angle ranges on gaze estimation accuracy.

**Experimental setup** We compared three rotation ranges for yaw, pitch, and roll axes:

- *Small rotation:* yaw $\in [-10°, 10°]$, pitch $\in [-10°, 10°]$, roll $\in [-5°, 5°]$.
- *Medium rotation:* yaw $\in [-20°, 20°]$, pitch $\in [-15°, 15°]$, roll $\in [-10°, 10°]$.
- *Large rotation:* yaw $\in [-40°, 40°]$, pitch $\in [-30°, 30°]$, roll $\in [-15°, 15°]$.

All other settings followed the main training configuration. We report angular gaze error (degrees) on TEyeD (Fuhl et al., 2021) and LPW (Tonsen et al., 2016) benchmarks.

**Discussion.** The results in Table 5 show that overly small perturbations fail to simulate diverse viewpoints, while excessively large rotations introduce unrealistic samples. Medium-range perturbations strike the best balance, improving cross-view robustness and cross-domain generalization.

## C.2   EFFECT OF THE KAPPA ANGLE BETWEEN THE OPTICAL AND VISUAL AXES.

The normalized optical axis $g$ is defined as the vector from the eyeball center $o_e$ to the iris center $o_i$, $g = \frac{o_i - o_e}{\|o_i - o_e\|}$. We consider g the approximated gaze vector. Note that we did not model the

Table 5: Effect of different rotation ranges in VCR on gaze estimation accuracy (angular error, degrees). Medium-range perturbations achieve the best trade-off.

| Rotation range | $D_{T_1} \to D_{T_2} \downarrow$ | $D_{T_1} \to D_S \downarrow$ |
|---|---|---|
| Small ($\pm 10°$ / $\pm 10°$ / $\pm 5°$) | 6.01 | 3.22 |
| Medium ($\pm 20°$ / $\pm 15°$ / $\pm 10°$) | **3.91** | **1.88** |
| Large ($\pm 40°$ / $\pm 30°$ / $\pm 15°$) | 4.65 | 2.79 |

Table 6: Quantitative results of individual training and testing of five subjects on TEyeD dataset. After eliminating the influence of kappa angle, the results show that applying more constraints is beneficial to improve the reconstruction accuracy of 3D eyeball model.

| Subject | Loss | TEyeD-subset_A | | | | |
|---|---|---|---|---|---|---|
| | | 3D gaze [°]↓ | 2D gaze [°]↓ | Sem. Iou | 2D pupil cent.[px]↓ | 2D eye cent.[px]↓ |
| subject1 | Gaze | 1.21 | 7.59 | N/A | N/A | N/A |
| | Sem. + Gaze + Cent. | 0.85 | 4.66 | 86.5% | 3.11 | 2.33 |
| subject2 | Gaze | 1.32 | 8.50 | N/A | N/A | N/A |
| | Sem. + Gaze + Cent. | 0.99 | 5.21 | 87.3% | 3.45 | 1.87 |
| subject3 | Gaze | 1.50 | 7.05 | N/A | N/A | N/A |
| | Sem. + Gaze + Cent. | 1.07 | 6.33 | 88.1% | 1.22 | 1.52 |
| subject4 | Gaze | 1.47 | 7.32 | N/A | N/A | N/A |
| | Sem. + Gaze + Cent. | 1.11 | 5.08 | 86.5% | 2.51 | 2.78 |
| subject5 | Gaze | 1.34 | 7.74 | N/A | N/A | N/A |
| | Sem. + Gaze + Cent. | 1.01 | 4.57 | 86.4% | 2.71 | 2.40 |

kappa angle offset between the optical and visual axes. In our previous experiments, we put more supervision on the whole eyeball, such as the center of the eyeball and pupil, as well as the edge of the projection. However, the accuracy has declined. To better understand its impact, we conducted controlled experiments. We trained and tested five subjects separately to eliminate the influence of kappa angle difference among different subjects. The experimental results in Tab 6 highlight that subject-dependent kappa offsets can negatively affect gaze estimation accuracy if ignored. Adding more subject-specific supervision constraints improves consistency of eyeball fitting and yields more reliable 3D gaze estimation.

### C.3 Sensitivity of MGR to 2D Edge Sparsity

**Motivation.** The Model-Guided Reconstruction (MGR) branch is supervised via sparse 2D edge points sampled on the pupil and iris contours. In practice, acquiring dense semantic labels can be expensive; therefore we analyze how the number of sampled 2D edge points $K$ affects reconstruction fidelity and gaze estimation performance. This experiment quantifies the trade-off between annotation cost (semantic sparsity) and final accuracy.

**Experimental setup.**

- **Backbone & training:** We use the same backbone and training hyperparameters as in the main paper (ResNet-18 / ResNet-50 variants as applicable). The training schedule, optimizer, weight decay and loss weights are kept identical to the main experiments to isolate the effect of $K$.

- **Point sampling:** For a given $K$ we uniformly sample $K_p$ contour points on the pupil and $K_i$ points on the iris such that $K = K_p + K_i$. By default we keep the pupil : iris ratio the same as in the main paper (e.g., $K_p : K_i = 4 : 1$ if the paper uses 128 pupil vs 32 iris).

- **K values tested:** $K \in \{8, 16, 32, 64, 128, 256\}$.

- **Datasets & evaluation:** Experiments are run on TEyeD-$D_{T_1}$ and evaluated on the same test splits used in the main paper. Metrics reported are 3D gaze angular error (degrees), 2D edge reprojection error (mean pixel distance), and semantic IoU.

**Losses and weights.** We keep the same overall objective as in the main paper:

$$\mathcal{L} = \lambda_g \mathcal{L}_{\text{gaze}} + \lambda_e \mathcal{L}_{\text{edge}} + \lambda_v \mathcal{L}_{\text{vcr}}.$$

When varying $K$ we do *not* change $\lambda_e$; this isolates the effect of the amount of 2D structural supervision. Reported runs keep $\lambda_g, \lambda_e, \lambda_v$ identical to the main experiments.

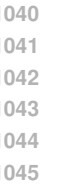
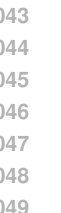
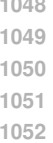

Figure 11: Sensitivity of the MGR branch to the number of 2D edge points $K$. Increasing $K$ rapidly reduces 3D gaze error up to $K = 128$, beyond which the performance saturates, indicating diminishing returns.

Table 7: Sensitivity of MGR to number of 2D edge points $K$. For each $K$ we report 3D gaze angular error (deg, lower better), 2D gaze angular error (deg, lower better), and semantic IoU (%, higher better).

| $K$ (total points) | 3D gaze (°) ↓ | 2D gaze (°) ↓ | Sem. IoU (%) ↑ |
|---|---|---|---|
| 8 | $2.47 \pm 0.08$ | $15.62 \pm 0.25$ | $78.5 \pm 0.6$ |
| 16 | $1.96 \pm 0.07$ | $13.24 \pm 0.21$ | $83.7 \pm 0.5$ |
| 32 | $1.64 \pm 0.05$ | $11.57 \pm 0.18$ | $87.2 \pm 0.4$ |
| 64 | $1.32 \pm 0.04$ | $9.87 \pm 0.15$ | $90.4 \pm 0.3$ |
| 128 | $\mathbf{1.11} \pm 0.03$ | $\mathbf{8.82} \pm 0.14$ | $\mathbf{92.1} \pm 0.3$ |
| 256 | $1.29 \pm 0.03$ | $9.75 \pm 0.13$ | $91.3 \pm 0.3$ |

**Concluding remark.** As shown in Fig. 11, this experiment empirically characterizes the annotation-performance trade-off for MGR and provides guidance for practical deployment: choose $K$=128 that yields near-saturated accuracy (the "knee"), which minimizes labeling cost while retaining reconstruction and gaze performance.

C.4    VISUALIZATION OF GAZE CONSISTENCY UNDER ROTATION

**Motivation** To further illustrate the effectiveness of our View-Consistent Regularization (VCR), we visualize predicted gaze vectors and eyeball projections under different synthetic rotations. This highlights how VCR enforces rotation-equivariant consistency in both appearance and structure.

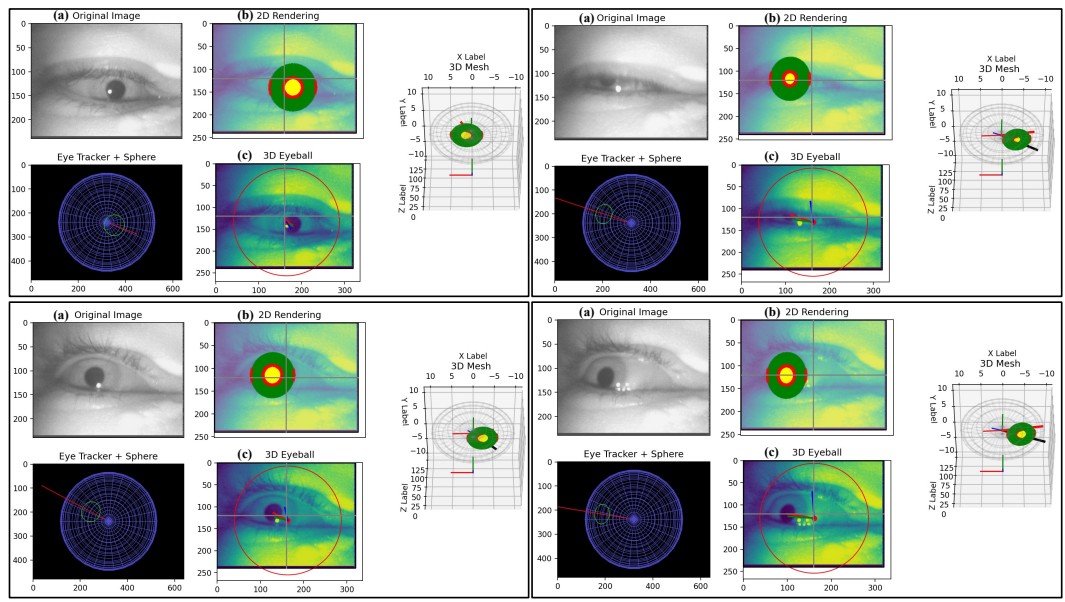

Figure 12: Visualization of gaze estimation under synthetic rotations. Each column shows (a) original image, (b) 2D rendering with VCR, and (c) rotated input with VCR. Red: ground-truth gaze; Blue: prediction without VCR; Green: prediction with VCR. With VCR, the predicted gaze aligns more consistently with the ground truth across different viewpoints, while the iris/pupil edges remain structurally faithful.

**Visualization setup** We apply yaw and pitch perturbations of $\pm 20°$ and project the reconstructed eyeball structures before and after rotation. For each case, we plot:

- Ground-truth gaze vector (red arrow).
- Predicted gaze vector without VCR (blue arrow).
- Predicted gaze vector with VCR (green arrow).
- Corresponding 2D iris/pupil edge projections (overlayed in the image).

**Discussion** As shown in Fig. 12, models trained without VCR are sensitive to viewpoint changes, causing gaze vectors to drift away from the ground truth and inconsistent iris contours. In contrast, VCR enforces consistent predictions under rotations, leading to both geometrically faithful eyeball reconstructions and improved cross-view gaze alignment.

# D  LIMITATION AND FUTURE WORK

## D.1  LIMITATIONS

Although SG-Gaze demonstrates strong accuracy and cross-domain generalization, several limitations remain that open promising directions for future work.

**(1) Subject-specific anatomical variation** Our framework currently approximates gaze by aligning the optical axis with the visual axis and does not explicitly model subject-dependent kappa angle offsets. As shown in our supplementary experiments, this simplification may introduce residual bias across individuals. Future work will investigate lightweight calibration strategies or personalized modules to better adapt to subject-specific anatomy.

**(2) Dependence on sparse 2D edge supervision** The Model-Guided Reconstruction (MGR) branch relies on weak 2D edge labels (pupil and iris contours). While our sensitivity analysis shows that even sparse labels are effective, annotation effort is still required. Extending MGR to leverage unsupervised geometric cues or self-supervised contour discovery could further reduce dependence on human annotation.

**(3) Synthetic-to-real domain gaps** Although the proposed View-Consistent Regularization (VCR) alleviates domain gaps, our training pipeline still relies on synthetic perturbations that may not capture the full diversity of real-world conditions (e.g., extreme illumination, occlusions, eyeglasses). Incorporating physics-based rendering, domain adaptation, or generative data augmentation may further improve robustness.

**(4) Deployment constraints** Our method has not yet been fully optimized for resource-constrained AR/VR headsets. Exploring lightweight backbones, pruning, or distillation will be essential to enable real-time gaze tracking on mobile or embedded platforms.

## D.2 FUTURE WORK

Building on SG-Gaze, several promising research directions can further align with the broader goals of the ICLR community:

**Subject-adaptive representation learning:** Develop lightweight calibration modules or meta-learning strategies to account for kappa angle offsets and anatomical variations, advancing personalized yet generalizable models.

**Self-supervised structural learning:** Move beyond annotated contours by exploiting self-supervised objectives and geometric consistency priors, enabling scalable training on large unlabeled datasets.

**Physics-aware domain adaptation:** Combine physically-grounded rendering, adversarial domain alignment, and generative augmentation to capture real-world shifts in illumination, occlusion, and device-specific imaging.

**Resource-efficient model design:** Pursue pruning, quantization, and distillation for low-latency deployment on AR/VR devices, bridging the gap between algorithmic advances and practical on-device applications.

**Multi-task and cross-modal extensions:** Explore joint learning with related tasks (e.g., iris recognition, user identification, eye-movement behavior analysis), and investigate integration with multimodal signals such as speech or head motion for richer human-centric modeling.

