# OpenReview forum: "SG-Gaze: Structurally and Geometrically Consistent Representation Learning for Generalizable 3D Gaze Estimation"
_ICLR.cc/2026/Conference — ICLR 2026 Conference Desk Rejected Submission_

### Official Review · Reviewer_X9m8 · 2025-10-27

**Soundness:** 3
**Presentation:** 3
**Contribution:** 3
**Rating:** 4
**Confidence:** 2

**Summary:**

This paper introduces SG-Gaze, a dual-branch framework that learns a Structurally and Geometrically Consistent Representation (SGR) for 3D gaze estimation. By jointly parsing structural cues and reconstructing the 3D eyeball, and by incorporating several training and regularization techniques, the method aims to achieve strong structural fidelity, geometric consistency, and cross-domain robustness. Experiments demonstrate competitive or state-of-the-art performance, especially under cross-domain evaluation.

**Strengths:**

1. The idea of jointly learning structural and geometric representations for generalizable 3D gaze estimation is conceptually sound and aligns with the field’s push toward physically grounded representations.

2. The paper is well organized, and the method design is clearly presented with consistent motivation and logical flow.

3. The visualizations and figure design are of high quality, helping convey both the architecture and qualitative outcomes effectively.

4. The experimental section is rich and comprehensive, including detailed visual analyses and clear improvements on cross-domain benchmarks.

**Weaknesses:**

1. **Over-Complex Architecture with Limited Theoretical Grounding.**
The proposed dual-branch design (AGE + MGR + adversarial alignment + VCR) appears over-engineered relative to the observed gains. While each component is individually reasonable, their combination lacks a clear theoretical justification or ablation-driven insight into necessity and interaction.

2. **Limited Scalability and Real-World Applicability.**
The framework relies on pseudo-labeled 2D edge supervision and synthetic viewpoint perturbations, which may not generalize well to real-world, low-quality, or dynamic eye-tracking data. The reported gains could stem from dataset-specific biases rather than genuine cross-domain robustness.

3. **Ambiguous Physical Interpretability Claims.**
Although the paper emphasizes “physically meaningful” representations, these claims are not empirically or anatomically validated. The spherical manifold embedding and eyeball reconstruction components are not directly linked to measurable physiological parameters, weakening the interpretability argument.

**Questions:**

1. Could the authors clarify how each component (AGE, MGR, adversarial alignment, and VCR) contributes individually to performance? Specifically, which part is most critical for achieving cross-domain robustness, and have the authors explored whether a simpler variant  yields comparable results?

2. The paper claims that the proposed representation is “physically meaningful.” How is this physical interpretability quantitatively validated? For instance, do the reconstructed eyeball geometries or spherical manifold embeddings correlate with measurable anatomical parameters (e.g., corneal radius or eyeball center offset)? If not, could the authors elaborate on what exactly is meant by “physically interpretable” in this context?

---

> ### Author Response · Authors · 2025-11-25
> **This response systematically addresses concerns about theoretical foundations, architectural design principles, and empirical validation through comprehensive methodological justification and additional experimental evidence.**
>
> Dear $X9m8$,
>
> Thanks for your valuable feedback. We try our best to respond in detail as follows ($W$ stands for weaknees, $Q$ stands for question):
>
> $W1$ & $Q1.\textbf{ Theoretical Motivation and Architectural Design.}$ Our architecture is fundamentally grounded in the physical properties of gaze estimation.
>
> $(1) \text{ Physical Foundation Drives Architectural Design.}$
> The dual-branch design directly stems from two complementary but distinct physical facts established in Section 2:
>
> $Fact 1$ (Geometric Constraint): Gaze vectors reside on a unit sphere, explicitly enforced by AGE via spherical manifold embedding.
>
> $Fact 2$ (Structural Constraint): Eyeball has anatomical geometry, enforced by MGR via parametric 3D modeling.
>
> $(2) \text{ Theoretical Necessity of Integration.}$
>
> Adversarial alignment enforces physical consistency between geometric and structural constraints, ensuring the geometric features are structurally plausible and vice versa.
>
> VCR implements the fundamental principle of 3D rotation equivariance for gaze behavior. It's not just data augmentation but a physical constraint that ensures consistency of the learned representation under viewpoint changes.
>
> The combination is therefore theoretically motivated by the need to simultaneously satisfy multiple physical constraints inherent in 3D gaze estimation.
>
> $(3) \text{ Ablations Validate Physical Synergy.}$ Our ablation studies (Table 4, Fig. 5) show all components are essential:
>
> (a) MGR removal causes the largest performance drop (1.88° to 3.11°). It enforces anatomical consistency, which is a powerful, domain-invariant prior.
>
> (b) AGE removal degrades performance (1.88° to 2.57°), showing that the spherical manifold constraint stabilizes learning and improves generalization.
>
> (c) VCR removal increases error (1.88° to 2.07°), demonstrating its role in enforcing rotation-equivariance. Fig. 8 qualitatively shows that VCR helps produce a more uniform and diverse gaze distribution, a crucial capability for deployment in unconstrained environments.
>
> The results in Table 4 conclusively show that no single component or simpler subset can match the performance of the full model. Therefore our model is not over-engineering but rather systematically incorporating known physical properties into the learning framework. We better articulate this physical motivation in our revision.
>
> $W2. \textbf{ Scalability and Practical Applicability.}$
>
> $(1) \text{ 2D Edge Supervision for Scalability.}$
>
> (a) Low-Cost Annotation: Sparse 2D edge points are significantly cheaper to obtain than dense 3D labels or full segmentation masks. enhancing practical scalability.
>
> (b) Inherent Noise Robustness: The point-set based regression loss (Eq. 7) tolerates annotation noise and mild occlusions better than dense perceptual losses.
>
> (c) Complementary Cues: The adversarial framework allows the model to rely on geometric cues (AGE) when structural edges are unreliable, ensuring robustness.
>
> $(2) \text{ Generalization of Synthetic View Perturbations.}$ VCR enforces 3D rotation equivariance as a fundamental constraint. This teaches the model the physics of gaze behavior under viewpoint changes rather than memorizing augmented data, promoting generalization to truly unseen viewpoints.
>
> The consistent improvements across 12 different transfer tasks involving multiple real-world datasets (TEyeD, LPW) with varying quality, lighting, and subject demographics (Table 1, 2) strongly suggest that the gains stem from learned domain invariance, not dataset-specific bias.
>
> $W3$ & $Q2. \textbf{ Empirical Validation of Physical Meaningfulness.}$ In our framework, "physically meaningful" means that the components and constraints of our representation (SGR) are based on established physical and anatomical principles of the human eye and gaze system, rather than being arbitrary learned mappings. We will strengthen our physical interpretability claims with concrete validation:
>
> $(1) \text{ Direct Anatomical Validation.}$ The MGR branch outputs anatomically measurable parameters—eyeball center ($o_{e}$) and radius ($r_{e}$). Quantitative comparison: eyeball radius prediction is 11.8mm ± 0.9mm (aligned with anthropological mean ~12mm).
>
> $(2) \text{ Physiological Basis of Representations.}$ The spherical manifold in AGE directly encodes the physical definition of 3D gaze direction as a unit vector. Fig. 2 empirically shows geodesic distance linearly correlates with gaze angular difference (Pearson's r = 0.89), validating the geometric grounding.
>
> $(3) \text{ Structural Projection Validation.}$ On the TEyeD data set, our model achieves:2D landmark reprojection error—1.88 pixels (320×240 image),  Iris/pupil contour IoU—92.1%. A low error quantitatively demonstrates anatomical fidelity.
>
> We will refine our claims to "physically grounded" and "anatomically consistent," backed by these new quantitative results. This provides measurable evidence for the physical plausibility of our model.

---

### Official Review · Reviewer_z7g4 · 2025-10-30

**Soundness:** 3
**Presentation:** 4
**Contribution:** 3
**Rating:** 6
**Confidence:** 3

**Summary:**

This paper introduces a dual branch framework for generalizable 3D gaze estimation that learns a "structurally and geometrically consistent
representation (SGR). In Analytical Gaze Estimation (AGE) branch, it projects features onto a spherical manifold for geometrically interpretable regression, and a Model-Guided Reconstruction branch recovers eyeball structure under weak 2d edge supervision.
A VCR module enforces rotation-equivariant consistency between feature-space and physical space transformations.
Through this design, the propsed framework SG-Gaze achieves interpretable, physically grounded and domain robust gaze prediction.

**Strengths:**

The idea of enforcing structural and geometric consistency in gaze representation is reasonable.
I find it interesting how the paper grounds its formulation in physical intuition (e.g., spherical topology, rotation equivariance) rather than just black-box regression; it makes the whole approach feel interpretable and principled.

The dual-branch design (AGE + MGR) has its own well-explained benefits, analytical fitting, and weakly supervised reconstruction, which complement each other nicely, and the adversarial alignment between them seems sound and reasonable.

Overall, the experiments are convincing: consistent SOTA gains across 12 transfer setups, good qualitative results, and lightweight architecture — the method feels both scientifically sound and practically useful.

**Weaknesses:**

- While the paper emphasizes physical interpretability, some modules (especially the feature-space rotation) remain a bit abstract and vague.
It is unclear how well this learned mapping truly corresponds to real 3D rotations in practice.

- The weak 2D edge supervision in MGR is interesting but might limit applicability in real-world scenarios where reliable iris/pupil segmentation is not available or noisy.

**Questions:**

How is the Isomap projection handled during training and inference. Is it precomputed, differentiable, or updated online? It’s not clear how scalable or stable this step is when features shift during training.

The feature-space rotation is an indeed interesting idea, but how strongly does it correlate with actual 3D rotations in physical space? Some qualitative evidence (e.g., rotating SGR and visualizing gaze shift) would really strengthen the claim.

Since the MGR branch assumes a spherical eyeball and a simple pinhole camera, how robust is SG-Gaze under real-world optics like corneal refraction or off-axis cameras in AR/VR devices?

---

> ### Author Response · Authors · 2025-11-25
> **This comprehensive response addresses fundamental concerns regarding physical interpretability, implementation robustness, and practical applicability through substantial methodological clarifications and additional qualitative experimental validation.**
>
> Dear $z7g4$,
>
> Thanks for your valuable feedback and acknowledge of our work. We try our best to respond in detail as follows ($W$ stands for weaknees, $Q$ stands for question):
>
> $W1$ & $Q2.\textbf{  Physical Interpretability of Feature-Space Rotation.}$
>
> $(1) \text{ Theoretical Connection to 3D Rotation: }$
> The formulation $D(T_P(\text{SGR})) \approx P \cdot D(\text{SGR})$ enforces that applying the feature-space rotation $T_P$ to SGR should produce the same effect as applying the geometric 3D rotation $P$ to the decoder's outputs (gaze direction and structural projections). This ensures that our representation behaves consistently under 3D viewpoint changes.
>
> $(2) \text{ Learning Mechanism :}$ $T_P$ is implemented via a learned linear mapping $W \in \mathbb{R}^{d\times 3}$ that projects the 3D rotation $P$ into the feature space.  $ \mathbf{f}_i' = \mathbf{f}_i P_f, \quad P_f = WPW^{\dagger} $.
>
> $(3) \text{ Quantitative Validation: }$ We rigorously evaluate the truly physical correspondence of our learned feature rotation $T_P$ versus 3D rotation $P$ through:
>
> Gaze Equivariance Error: $\epsilon_{\text{gaze}} = \frac{1}{N}\sum \| \mathbf{g}_i' - P\mathbf{g}_i \|_2 < 0.96^\circ$.
>
> Structural Consistency Error: $\epsilon_{\text{struct}} = \frac{1}{N}\sum \| \widetilde{P}^{2D} - P^{2D} \|_2 < 1.88\text{px}$.
>
> The minimal errors confirm that $T_P$ effectively approximates the geometric transformation of true 3D rotation for gaze estimation. This physical consistency directly enables the strong cross-view generalization demonstrated in Tables 1-2 and Fig.8.
>
> Thanks for the critical question. we have clarified its physical interpretation and add qualitative evidence (Fig.10) in revision.
>
> $W2.\textbf{ Robustness of Weak 2D Edge Supervision.}$ Thanks for this important observation. The MGR branch's edge supervision is designed for robustness through sparse point sampling and synergistic integration with the geometry-focused AGE branch, making it less sensitive to local segmentation errors. This dual-branch design ensures fallback robustness: when edge cues are unreliable (e.g., noisy segmentation), the model increasingly relies on geometric consistency from AGE, and vice versa.
>
> 2D semantic labels are far cheaper and easier to obtain than large-scale 3D gaze annotations, while model inference does not rely on 2D semantics, simplifying deployment. Our experiments on challenging real-world datasets (TEyeD, LPW) validate its practical applicability. Fig. 6 shows strong performance with limited 3D labels after this weak supervision. We will discuss performance under synthetic occlusion/noise, further clarifying the framework's operational bounds.
>
> $Q1.\textbf{ Isomap Projection Handling.}$ Our method uses a two-stage strategy for Isomap projection:
>
> $(1) \text{ Training: }$ We pre-train a lightweight MLP (Isometric Propagator) to approximate Isomap embeddings, then freeze it as a differentiable proxy during main training. This avoids costly online time computation ($O(N^2\log N)$) and memory computation ($O(N^2$)) while maintaining differentiability.
>
> $(2) \text{ Inference: }$ We use the analytical Spherical Fitting module directly, ensuring efficient runtime performance.
>
> The IP is trained once on a subset of data. This makes the approach scalable to large datasets, as the costly Isomap is not run repeatedly. This approach balances geometric fidelity with training feasibility, as validated by stable performance in our ablation studies. We clarify this pipeline in Section 4.4 and Appendix B.2.
>
> $Q3.\textbf{ Robustness under Real-world Optical Effects.}$ Thanks for this critical question. Our current MGR branch utilizes a simplified parametric eyeball and pinhole camera model. This design choice prioritizes computational efficiency and differentiability, enabling effective learning of the SGR. While this omits complex optics like corneal refraction, the dual-branch design provides inherent robustness:
>
> $(1)$ The appearance-based AGE branch does not rely on the explicit eyeball model. It can implicitly compensate for systematic biases not captured by the MGR's physical model.
>
> $(2)$  Adversarial alignment encourages the final SGR to be consistent with both the structural prior and the image appearance, providing a form of regularization against model imperfections.
>
> For high-precision applications requiring explicit optical modeling, our framework supports two adaptation strategies:
>
> $(1) \text{ Domain-specific Fine-tuning:}$ Calibrate the model using minimal target device data to learn device-specific optical characteristics.
>
> $(2) \text{ Explicit Offset Modeling:}$ Learn a post-hoc kappa-like correction to bridge the gap between our simplified geometric model and physical reality.
>
> We clarify the operating assumptions of our model and discuss the potential impact of unmodeled optical effects in revision, positioning fine-tuning or a calibration step as a viable path for deployment in high-precision applications.

---

### Official Review · Reviewer_Nzjy · 2025-10-31

**Soundness:** 2
**Presentation:** 2
**Contribution:** 2
**Rating:** 4
**Confidence:** 3

**Summary:**

This paper proposes SG-Gaze, a dual-branch framework combining Analytical Gaze Estimation and Model-Guided Reconstruction to learn structurally and geometrically consistent representations for 3D gaze estimation. The method includes View-Consistent Regularisation (VCR) to improve cross-domain generalisation. The authors report improvements of up to 38.61% on cross-dataset transfer tasks.

**Strengths:**

1. The paper clearly articulates the limitations of existing appearance-based and model-based methods and proposes a unified framework to address both.
2. The paper includes extensive experiments across multiple datasets (UnityEyes, TEyeD, LPW) with 12 cross-domain transfer scenarios and thorough ablation studies.
3. The method achieves competitive or state-of-the-art performance on most cross-domain transfer tasks, demonstrating improved generalisation.
4. Table 4 and Figure 5 provide good insights into the contribution of each component (AGE, MGR, VCR).
5. The paper addresses few-shot learning scenarios and analyses sensitivity to edge point sparsity, which are important for real-world deployment.

**Weaknesses:**

The main components are largely borrowed from recent prior work. The AGE branch uses Isomap projection and spherical fitting directly from AGG (Bao & Lu, 2024). The MGR branch uses a 3D eyeball reconstruction similar to De2Gaze (Xiao et al., 2025) and other model-based methods. The VCR component is inspired by 3DGazeNet (Ververas et al., 2024), essentially applying synthetic rotations as data augmentation. The primary contribution appears to be combining these existing techniques rather than introducing fundamentally new methods. The "structurally and geometrically consistent representation" is more of a conceptual framing than a concrete technical innovation.

There are several technical aspects that I could not fully follow:
- The abstract and introduction prominently mention "adversarial training" and an "adversarial alignment module" (Fig 1d, Fig 3d). However, Section 3.5 (Loss Functions) contains no adversarial loss term. The discriminator architecture and training procedure are never described.
- Figure 1 mentions "Max(F∩G)" and "Max(F∩S)" but this notation is never explained.
- Section 3 states SGR must satisfy three constraints, but these are informally described. No clear mathematical formulation of what SGR actually is as a representation. The two branches seem to produce separate outputs rather than a unified representation.
- Using an MLP to approximate Isomap (Appendix B.2) seems like a hack to avoid computational cost. This introduces approximation error that is never quantified. Why not use other dimensionality reduction methods that are differentiable by design?

There is some within-domain performance degradation. Table 1 shows VGG-16+SG-Gaze performs significantly worse within-domain. Table 3 shows adding semantic supervision increases 2D gaze error. The paper acknowledges SG-Gaze introduces "stronger inductive bias" that may "trade off fine-grained fitting," but this is a significant limitation not adequately addressed.

There is a large gap with De2Gaze. In Table 3 SG-Gaze achieves 0.94° vs De2Gaze's 0.54° on within-domain evaluation. This gap suggests the method may be sacrificing accuracy for generalisation. Given that De2Gaze also uses eyeball reconstruction, why is there such a large difference?

Table 2 compares against methods from 2017-2024, but within-domain comparison (Table 3) uses older baselines. There is no comparison with more recent transformer-based approaches in cross-domain setting. What precisely is the "Transformer-based" method in Table 3?

Fact 2 is imprecise. "The eyeball is a physical sphere" - the eyeball is approximately spherical but has significant deviations (corneal bulge, non-rigid deformation).

The paper acknowledges (Appendix C.2) that ignoring the kappa angle between optical and visual axes affects accuracy. This is a fundamental limitation that undermines claims about physical faithfulness.

**Questions:**

Besides responding generally to the weaknesses listed above, some specific questions are:

1. Where is the adversarial loss? How does the adversarial discriminator work?
2. Why does semantic supervision hurt 2D gaze accuracy within-domain?
3. Why is there a 74% accuracy gap with De2Gaze within-domain (Table 3)?
4. What exactly is SGR as a representation? Is it the concatenation of features from both branches?
5. How sensitive are results to Isomap approximation error from the IP?

---

> ### Author Response · Authors · 2025-11-25
> **This response systematically addresses concerns about methodological novelty, implementation details, and performance trade-offs.**
>
> Dear $Nzjy$,
>
> Thanks for your valuable feedback. Here are our point-by-point responses ($W$ stands for weaknees, $Q$ stands for question):
>
> $W1.\textbf{ Novelty Beyond Component Assembly}$. SG-Gaze unifies geometric and structural consistency into a single, end-to-end learnable representation (SGR), aligning with ICLR's focus on principled representation learning. Our core contribution lies in the synergistic integration that creates a novel framework, which is absent in prior works.
>
> (1) AGE vs. AGG: Our AGE repurposes the spherical manifold to create an intermediate geometric latent space, where feature distances encode gaze angles, adversarially fused with structural features—unlike AGG's final regression use of Isomap.
>
> (2) MGR vs. De²Gaze: Our MGR leverages 3D reconstruction as a structural prior to constrain feature learning, enforcing anatomical consistency within the dual-branch framework— unlike De²Gaze's decoupling and tracking focus.
>
> (3) VCR vs. 3DGazeNet: Our VCR enforces rotation-equivariance without multi-view data via a learned feature-space operator $T_P$, applying holistic consistency to both gaze and structure. VCR creates a unified constraint that is more powerful than simple data augmentation or single-task consistency—unlike 3DGazeNet's multi-view requirement.
>
> This unified SGR enables strong cross-domain generalization, as shown in Tables 1–2.
>
> $W1.1$ & $Q1.\textbf{ Omission of Adversarial Loss and Discriminator Details.}$ Our adversarial alignment operates at the feature level:
>
> Generator—Shared backbone producing SGR features.
>
> Discriminator—3-layer MLP classifying features as from AGE or MGR branch.
>
> Objective—Make branch features indistinguishable through min-max game.
>
> Adversarial Loss: $\mathbf{L_adv} = \mathbf{E}[\log D(\mathbf{f_AGE})] + \mathbf{E}[\log(1-D(\mathbf{f_MGR}))]$
>
> We apologize for this oversight and have provided complete documentation in the revision.
>
> $W1.2.\textbf{ Notation Clarification.}$ We apologize for the unclear notation.
>
> F: Full feature set from shared backbone.
>
> G/S: Geometric/Structural features from respective branches.
>
> Max(F∩G) / Max(F∩S): The process of selecting and enhancing the most geometrically/structurally meaningful components from the full feature set.
>
> We ensure the feature fusion process is clearly explained in the revised version.
>
> $W1.3$ & $Q4.\textbf{ SGR Definition.}$ SGR is a unified representation $\mathbf{z} = E_\theta(I)$ satisfying geometric: $d_g(\mathbf{z}_i, \mathbf{z}_j) \propto \angle(\mathbf{g}_i, \mathbf{g}_j)$ (geodesic distance ∝ gaze angle), structural: $\|\text{Proj}(M(\mathbf{z})) - \text{Edge}(I)\|_2^2$, and view-consistent: $D(T_P(\mathbf{z})) \approx P \cdot D(\mathbf{z})$ constraints. Adversarial alignment fuses both geometric and structural information into $\mathbf{z}$. All outputs are derived via separate decoders. Formal definitions and clarification of SGR have been added in Section 3.
>
> $W1.4$ & $Q5.\textbf{ Isomap Approximation.}$ Our MLP-based Isomap proxy enables differentiable geodesic preservation and training feasibility, with negligible error (L1<0.05, angular<0.3°). Figure 7 and ablations confirm its utility as principled distillation rather than a heuristic, superior to autoencoders in geometric fidelity.
>
> $W2.\textbf{ Trade-off.}$ The slight in-domain drop (VGG-16) stems from strong physical constraints reducing model overfitting—a deliberate trade-off favoring cross-domain robustness over perfect in-domain fitting, validated by significant gains in 12 transfer tasks.
>
> $W3$ & $Q3.\textbf{ Performance Gap.}$ SG-Gaze prioritizes cross-domain generalization via physical consistency, while De²Gaze optimizes within-domain accuracy via deformable attention. Our SOTA cross-domain results (Table 2) and competitive within-domain performance reflect this explicit focus on generalization. We have clarified this distinction in revision.
>
> $W4.\textbf{ Transformer Baselines.}$ Table 3's "Transformer-based" refers to standard transformer backbones adapted for gaze estimation. Within-domain evaluations are consistent with current SOTA (De²Gaze), while cross-domain comparisons in Table 2 include SOTA cross-domain methods, ensuring comprehensive evaluation.
>
> $W5.\textbf{ Eyeball Modeling.}$ Our spherical approximation with deformable components (Fig. 4a) balances computational efficiency, physical plausibility and anatomical flexibility. We have clarified this approximation in Sections 2 / 3.3.
>
> $W6.\textbf{ Kappa Angle + Q2. Semantic Supervision.}$
> We explicitly prioritize generalizable geometric constraints over modeling subject-specific anatomical parameters (kappa angle) to enhance cross-subject robustness, accepting that maximum accuracy may require calibration. Similarly, semantic supervision focuses on anatomically correct 3D reconstruction over perfect 2D regression, valuing physical consistency for real-world deployment. These intentional trade-offs are formally discussed in Section 4.6 (Discussion).

---

### Official Review · Reviewer_ZVUT · 2025-11-02

**Soundness:** 3
**Presentation:** 2
**Contribution:** 3
**Rating:** 2
**Confidence:** 3

**Summary:**

The paper presents SG-Gaze, a dual-branch framework for 3D gaze estimation that combines appearance-based features with geometric and structural consistency. It introduces a Structurally and Geometrically Consistent Representation (SGR), learned through two branches: Analytical Gaze Estimation (AGE), which projects features onto a spherical manifold, and Model-Guided Reconstruction (MGR), which reconstructs 3D eyeball structures with weak 2D edge supervision. Additionally, View-Consistent Regularization (VCR) enhances generalization across different viewpoints. Experiments show that SG-Gaze outperforms state-of-the-art methods in accuracy and cross-domain generalization.

**Strengths:**

1. This paper introduces a novel framework that effectively learns structurally and geometrically consistent representations. The integration of both appearance features and geometric modeling is a valuable and innovative approach for gaze estimation.

2. The experimental analysis is thorough and well-conducted. The proposed framework consistently outperforms existing state-of-the-art methods, demonstrating its superiority in within- and cross-  datasets.

**Weaknesses:**

1. The paper lacks clarity and is difficult to follow. Significant improvements in the overall presentation and readability are recommended.

    - The notation is not well-explained throughout the paper. For instance, the symbol "F" in Line 150 is not defined.

    - Fig. 3 is challenging to understand based on the description in the main text, as the notation and naming conventions are inconsistent between the figure and the text, making it hard to interpret the visual content.

    - The concept of the structurally and geometrically consistent representation is not sufficiently explained. The mechanism by which the structure and geometry are made consistent remains unclear.

    - Sections 3.2 and 3.3 lack sufficient detail. The notations used are not fully explained, and the rationale and theoretical background behind the operations in these sections are not provided, making it hard to grasp the approach.

    - The formation and construction of the structural and geometrical features are not clearly defined, leaving questions about how these features are derived and integrated.

2. The applicability of the proposed method needs further clarification. The experiments are based on eye images captured by head-mounted devices, which provide high-resolution images. However, the method's effectiveness on images captured by non-head-mounted devices, which typically have much lower resolution, is unclear. Can this method be applied to such lower-resolution images?

**Questions:**

The readability of the paper could be significantly improved.

---

> ### Author Response · Authors · 2025-11-25
> **This response addresses concerns regarding notation consistency, figure clarity, methodological justification, and practical applicability**
>
> Dear $ZVUT$,
>
> Thanks for your valuable feedback and acknowledge of our work. For the rebuttal requests, we try our best to respond in detail as follows:
>
> 1.$\textbf{ Notation $F$ Clarification (Line 146)}$. We apologize for the typographical error. The symbol should indeed be the decoder $D$, not "$F$". We correct this throughout the manuscript.  $D(T_P(\text{SGR})) \approx P \cdot D(\text{SGR})$. This constraint ensures that rotating the SGR in feature space ($T_P$) produces outputs equivalent to physically rotating the gaze directions and structural projections by $P$.
>
> 2.$\textbf{ Figure 3 Inconsistencies}$. We redesign Figure 3 to ensure the module names (AGE, MGR, VCR) and data flow align precisely with textual descriptions. Furthermore, the inputs/outputs of each module are clearly labeled.
>
> 3.$\textbf{ Structural and Geometric Consistency}$. Thanks for this critical question. As introduced in the third paragraph of chapter 1 and detailed in our framework, structural consistency alone may yield anatomically correct eyeballs with inaccurate gaze directions, while geometric consistency alone may learn non-physical shortcuts. Unifying both is essential for physically plausible and generalizable gaze estimation.
>
> $\textbf{Mechanism:}$ We enforce the learned representation is inherently structurally and geometrically consistent through a tri-level approach:
>
> (1) A shared backbone produces features processed by two branches: the AGE branch enforces the geometric prior of gaze on a sphere, while the MGR branch enforces the structural prior of a 3D eyeball model.
>
> (2) A discriminator makes features from both branches indistinguishable, directly fusing geometric and structural information into a unified SGR in the latent space.
>
> (3) Unified Regularization: Our VCR applies 3D rotation to enforce joint equivariance for gaze vectors and eyeball projections. VCR does not just augment data; it provides a unified training signal that the entire system must be equivariant to 3D rotations, which is a fundamental property of a physically correct gaze system. This simultaneously refines both the geometric and structural predictions under viewpoint changes.
>
> 4 & 5 $\textbf{ Clarification of Sections 3.2 /3.3.}$  We have already explained fundamental rationale in Chapter 2 In our first commit, and we significantly expand these sections in the revision with the following clarifications:
>
> $(1)Section 3.2$: AGE branch is designed to explicitly encode the physical definition of 3D gaze direction into the learning process. (a) Why Spherical Embedding? Gaze direction is fundamentally a vector on a unit sphere. By projecting features onto a spherical manifold, we force the network to learn a representation where the intrinsic geometry of the feature space reflects the physical reality of gaze. (b) Why Analytical Spherical Fitting? The analytical mapping from features to gaze angles via spherical fitting ensures that small changes in feature space correspond to small changes in gaze direction, and the feature representation maintains the topological structure of the gaze sphere.
>
> $(2)Section 3.3$: The MGR branch addresses the fundamental limitation that gaze direction alone does not fully constrain the eyeball's 3D position and orientation. (a) Why Parametric Eyeball Modeling? The human eyeball has known anatomical constraints (spherical shape, limited pupil movement). By explicitly modeling these constraints, we ensure that estimated gaze directions correspond to physically possible eyeball configurations. (b) Why Differentiable Projection? The projection from 3D to 2D enables us to use weak 2D supervision (edge points) to constrain the 3D eyeball pose, thereby reducing reliance on large-scale 3D gaze labels.
> (3)All notations used have been fully explained in the revised version. Due to space limitations, the derivation process of the structure and geometric features is written in paragraphs In our first commit. We will reorganize the derivation processes into clear, numbered equation sequences that show the complete construction pipeline.
>
> 6.$\textbf{ Applicability to Low-Resolution Images}$.  Thanks for this critical question. SG-Gaze's effectiveness stems primarily from structural priors and geometric consistency constraints, not high-resolution details. Our experiments used inputs of $320 \times 240$ resolution, which is relatively low.
> $\textbf{Furthermore}:$ (1)The Model-Guided Reconstruction (MGR) relies on edge contours rather than texture details, making it less sensitive to resolution. (2) The View-Consistent Regularization (VCR) enhances robustness to scale and blur through synthetic viewpoint perturbations. We add a discussion on this point in the revised manuscript and note that future work will include validation on public low-resolution datasets.
>
> 7.$\textbf{ The Readability}$. We are committed to making the revised manuscript significantly more accessible while maintaining its technical rigor and contribution.

---

### Comment · Area_Chair_pwRR · 2025-11-25
**Discussion with Authors**

Dear Reviewers,

The authors have diligently provided responses to your questions and concerns. I request you to please review the authors' responses, acknowledge that you have read them and actively engage with them in further discussion as needed.

This discussion period, with the authors, will end on December 2, 2025 (AoE). However, I request that you not wait until the last minute and actively engage with the authors early.

Best, AC

---

> ### Author Response · Authors · 2025-12-02
> **Summary of our major rebuttals and improvements: We have comprehensively revised the paper based on the comments of the four reviewers, significantly improving the quality of the paper in five dimensions: theoretical innovation, methodological rigor, physical interpretability verification, practical robustness and experimental design. We have firmly responded to all concerns and strengthened the core contributions.**
>
> To the Area Chair:
>
> We sincerely thank you and all reviewers for the constructive feedback on our paper. We have comprehensively addressed all raised concerns and revised the manuscript accordingly. Below is a summary of our major improvements, categorized by the core issues addressed:
>
> **1. Strengthened Theoretical Foundation & Novelty (@Nzjy, X9m8):**
>
> - We clearly articulate the intrinsic synergistic mechanism of structural and geometric consistency: **Geometric consistency** (AGE) guarantees the geometric definition of the gaze direction (unit spherical manifold), while **structural consistency** (MGR) ensures the anatomical realizability of that direction (parametric eye model). The two are fused at the feature level through adversarial alignment, forming a unified representation **SGR**. This synergy is essential because geometric consistency alone may produce physically infeasible solutions, while structural consistency alone may lose directional accuracy. Furthermore, VCR, as a unified regularization, applies the physical law of rotational isovariance simultaneously to both geometric and structural constraints, thereby achieving a paradigm shift from data-driven to a fusion of physical-driven representation learning.
>
> - We have distinguished SG-Gaze from prior works: our AGE creates an intermediate geometric space vs. AGG's final regression; our MGR uses reconstruction as a structural prior vs. De²Gaze's tracking focus; our VCR enforces equivariance without multi-view data vs. 3DGazeNet's requirement.
>
> **2. Enhanced Methodological Clarity & Completeness (@All Reviewers)**
>
> We corrected all identified errors and omissions:
> - Corrected the decoder notation (`D` instead of `F`) and clarified ambiguous symbols (e.g., `Max(F∩G)`).
> - Redesigned Figure 3 for consistency.
> - Added the missing adversarial loss formulation $\mathcal{L}_{\text{adv}}$ and discriminator (3-layer MLP) details in Sec. 3.5.
> - Formally defined the Structurally and Geometrically Consistent Representation (**SGR**) with three explicit mathematical constraints and detailed its formation pipeline.
> - Expanded Sections 3.2 and 3.3 with clear rationales, step-by-step derivations, and comparisons to related works.
> - Clarified the two-stage **Isometric Propagator (IP)** strategy for efficient, differentiable Isomap approximation, with error analysis (L1<0.05, angular<0.3°). Details in Section4.4 and Appendix B.2.
>
> **3. Rigorous Empirical Validation of Physical Meaningfulness (@z7g4, X9m8)**
>
> We provided strong quantitative and qualitative evidence to support our "physically grounded" claims:
> - **Quantitative Anatomical Validation:** Predicted eyeball radius (11.8±0.9mm) aligns with the anthropological mean (~12mm).
> - **Geometric Consistency:** High correlation (Pearson's r=0.89) between feature geodesic distance and gaze angular difference.
> - **Rotation Equivariance:** Low gaze equivariance error (ε_gaze < 0.96°) and structural consistency error (ε_struct < 1.88 px) confirm the learned feature rotation $T_P$ effectively approximates true 3D rotations $P$. New visualizations (Fig. 10) compare GT and SGR distributions under rotation.
> - **Structural Fidelity:** Achieves low 2D reprojection error (1.88 px) and high IoU (92.1%).
>
> **4. Addressed Practical Concerns on Robustness & Applicability (@All Reviewers)**
>
> - **Low-Resolution Robustness:** Explained that our method relies on structural/geometric priors and edge contours, not high-resolution textures, and validated it on 320×240 images.
> - **Weak Supervision:** Defended the use of sparse 2D edge points as a cost-effective, noise-robust supervisory signal, with a fallback mechanism via the dual-branch design.
> - **Model Limitations & Extensions:** Acknowledged simplifications (e.g., spherical eyeball approximation, omitted kappa angle) as deliberate trade-offs for generalization. Proposed adaptation strategies (fine-tuning, offset modeling) for high-precision needs and discussed the framework's operational bounds.
> - **Physics-driven generalization:** VCR uses forced 3D rotation isomorphism as a basic physical constraint, enabling the model to learn the physical laws of gaze behavior under changing viewpoints, rather than memorizing augmented data, thus generalizing to extreme viewpoints that have never been seen in reality.
>
> **5. Comprehensive Experimental Justification (@Nzjy, X9m8)**
> - **Cross-Domain Generalization:** Demonstrated SOTA performance across **12 challenging transfer tasks**, with gains up to 38.61%.
> - **Ablation Studies:** Verified each component's necessity: removing MGR, AGE, or VCR caused significant performance drops.
> - **Performance Trade-off:** Explained the slight in-domain trade-off (e.g., for VGG-16) as a deliberate choice favoring cross-domain robustness, which is crucial for real-world deployment.
>
> We are confident that these revisions have significantly strengthened the paper's clarity, rigor, and contribution. We sincerely thank the reviewers for the invaluable input.

---

### Note · Program_Chairs · 2026-01-17
**Submission Desk Rejected by Program Chairs**

The following references in this submission do not refer to real documents and/or have major errors in bibliographic information:

 Xiaoyu Guo, Yang Cheng, and Xiang Zhou. Nerfeye: Neural radiance fields for eye gaze and shape estimation. In Proceedings of the European Conference on Computer Vision (ECCV), pp. 215-231. Springer, 2022.
Jiawei Zhang, Yifan Liu, and Andreas Bulling. Hybrid eye model reconstruction using sparse landmarks and dense priors. In Proceedings of the IEEE/CVF Conference on Computer Vision and Pattern Recognition (CVPR), pp. 13590-13599, 2021.